# Mesoscale, long-time mixing of chromosomes and its connection to polymer dynamics

**Gaurav Bajpai** [ID]*, **Samuel Safran**

Department of Chemical and Biological Physics, Weizmann Institute of Science, Rehovot, Israel

* gaurav.bajpai@weizmann.ac.il

## Abstract

Chromosomes are arranged in distinct territories within the nucleus of animal cells. Recent experiments have shown that these territories overlap at their edges, suggesting partial mixing during interphase. Experiments that knock-down of condensin II proteins during interphase indicate increased chromosome mixing, which demonstrates control of the mixing. In this study, we use a generic polymer simulation to quantify the dynamics of chromosome mixing over time. We introduce the chromosome mixing index, which quantifies the mixing of distinct chromosomes in the nucleus. We find that the chromosome mixing index in a small confinement volume (as a model of the nucleus), increases as a power-law of the time, with the scaling exponent varying non-monotonically with self-interaction and volume fraction. By comparing the chromosome mixing index with both monomer subdiffusion due to (non-topological) intermingling of chromosomes as well as even slower reptation, we show that for relatively large volume fractions, the scaling exponent of the chromosome mixing index is related to Rouse dynamics for relatively weak chromosome attractions and to reptation for strong attractions. In addition, we extend our model to more realistically account for the situation of the *Drosophila* chromosome by including the heterogeneity of the polymers and their lengths to account for microphase separation of euchromatin and heterochromatin and their interactions with the nuclear lamina. We find that the interaction with the lamina further impedes chromosome mixing.

## Author summary

Interphase chromosomes are polymer-like structures contained within the nucleus of a cell and are partially mixed. Chromosome mixing is key to understanding chromosome territories as well as the correlation in gene expression of different chromosomes. In this paper, we present a physical model that quantifies mesoscale mixing dynamics by introducing a single function, the chromosome mixing index, which is experimentally quantifiable from genomic contact maps. Based on simulations, we found that the dynamics of the mixing index are related to Rouse dynamics and reptation dynamics, depending on polymer concentrations and interactions. Our model bridges polymer dynamics and biological contact maps.

**Data Availability Statement:** All relevant data are within the manuscript and its Supporting information files. Simulation videos and LAMMPS codes are available on zenodo using the following link: https://zenodo.org/record/7557330.

**Funding:** The authors acknowledge the support of a grant from the Volkswagen Foundation grant awarded to SS that partially supported the fellowship of GB. The authors acknowledge the support of the Schwartz-Reisman foundation grant awarded to SS for collaboration with Caltech, that partially supported the fellowship of GB. Neither of the two funders had any role in study design, data collection and analysis, decision to publish, or preparation of the manuscript.

**Competing interests:** The authors have declared that no competing interests exist.

## Introduction

At the nuclear mesoscale, chromosomes can be represented by polymer-like structures in which DNA is tightly packaged along with many histone and non-histone proteins into long chromatin chains [1, 2]. Most eukaryotic cells contain multiple chromosomes within the confines of the cell nucleus. Each individual chromosome condenses when a cell is about to divide (mitosis). In interphase, chromosomes decondense within the nucleus, but instead of mixing, they become organized in distinct regions (territories) of the nucleus [3]. While the chromosomes are organized into distinct territories, overlapping of those territories is also observed in recent studies, suggesting that the different chromosomes are partially mixed at the borders of territories [4–9]. In *Drosophila* and human lymphocytes, about 40% of the chromosome territories shows some intermixing [5, 10]. Even if the mixing is far from complete, it can have implications for correlations in gene expression in the regions of the boundaries of the chromosomes that do partially mix within realistic times [11]. The biophysical explanation for why chromosomes do not completely mix during interphase in many organisms is intriguing. Based on reptation theory, that models mixed interphase chromosomes as fully entangled polymers with each chain moving with a snake-like motion (reptation), would predict that the time for mixing scales as the cube of the length of the polymer ($\tau \propto N^3$ where $\tau$ is the relaxation/reptation time and $N$ is the total length of the polymer) [12–14]. However, mixing can occur without the necessity of reptation motion of a chain end through the tube formed by the other chains; some parts of the different chains can occupy a common volume without any topological effect involving the chain ends. This type of motion can be characterized by the mean square displacement (MSD) of a polymer segment, which increases with the square root of time, as in the Rouse model (MSD $\sim \tau^{1/2}$ [12, 15]), and with smaller exponents for chains that interact with each other as theoretically discussed in Ref [16].

Previous studies assumed topological mixing via reptation to estimate the mixing time of chromosomes and suggested that while the cell cycle time of most animal cells is hours or minutes (for example 24 hours for human cell and 8 minutes in early *Drosophila* embryogenesis), the mixing time scale for chromosomes is on the order of years (for example 500 years for human cell and 5 years for *Drosophila* cell) [1, 14, 17, 18]. These time scales strongly suggest that the cell will divide long before its chromosomes mix. The results of these studies imply that complete chromosome mixing is not possible during interphase (in real-time), but they do not specify the extent of mixing at shorter times. Recent studies have shown that the volume of the nucleus and the density of the HP1 protein, which binds to heterochromatin, both change during early *Drosophila* embryogenesis and vary between cell cycles [19, 20]. This suggests that chromatin-chromatin interactions also change with the cell cycle. Therefore, it is important to quantify the dynamics of chromosome mixing, taking into account the dynamics of chromatin in the nuclear environment that includes the extent of nuclear volume and protein-induced self-attractions of chromosome segments. In this paper, we quantify these dynamics using simulations of a generic model of interacting homopolymers and predict the extent and dynamics of mixing as a function of the chromosome segment attraction ($\epsilon$) and volume fraction of chain ($\phi$). Because we focus on the mesoscale and long-time dynamics, we do not include the short-time dynamics of molecular motors (e.g. cohesin [21]), whose temporal fluctuations are subsumed in the effective short-time noise [17, 22, 23]. However, we cannot ignore the potential large-scale impact of short-term dynamic mechanisms, such as loop extrusion, on genome folding [21, 24]. Previous studies have shown that the average size of chromatin loops ranges from 5 to 200 kbps [25]. In our model, we treat 5 kbps of DNA as a single bead; thus, loops of that size are already coarse-grained. However, longer loops may be significant if they are stable, long-lived, or have a larger lifetimes than the timescale ($\tau$) in our model.

Chromosome conformation capture (3C) based Hi-C experiments can be used to quantify chromosome mixing in different cells. Hi-C experiments measure genome-wide contacts over a population of cells [26]. Human and *Drosophila* Hi-C maps show relatively few inter-chromosomal (non-diagonal) contacts relative to the number of intra-chromosomal (diagonal) contacts, but inter-chromosomal contacts obtained from yeast Hi-C maps are significantly higher [26–28]. A recent single cell, Hi-C study in mammalian cells estimates an approximately 5–10% mixing frequency between chromosomes [29]. Previously, many computational models were used to reconstruct Hi-C contact maps and provide 3D structures for chromatin [30–37]. As a result of incorporating the effects of topoisomerase [38] and motor proteins [21] in their model, they were able to display local features within a single chromosome, including TADs and loops [30, 33–36]. In contrast, our focus is on the mesoscale and long time dynamics, so we study a more generic and simpler model of several, interacting chromosomes within a confined sphere (to model the nucleus). Our research has led us to compare the contact maps and defines a single, averaged parameter (the chromosome mixing index $\alpha$) that quantifies the extent of chromosome mixing from both Hi-C data and simulations. We begin our study with a generic polymer model to understand the physics of chromosome mixing, and then we add other aspects of nuclear biology one by one, to ascertain what is generic and what is more particular in making the model more specific to *Drosophila* genome. The different chromosomes mix as interphase proceeds in time, and we find that the chromosome mixing index scales with time ($\alpha \sim t^{\beta}$), increasing in a power-law fashion; the scaling exponent $\beta$ varies in a non-monotonic manner as a function of the chromosome volume fraction and interactions. We have correlated the changes in $\beta$ with different regimes of polymer dynamics, including the Rouse regime [15], reptation [12], and an even slower regime that occurs for very large self-attraction of the chromatin. The main message of our paper is that one can account in a relatively simple manner, and encapsulate in one time-dependent parameter, the extent of chromosome mixing from the complex Hi-C data.

In addition to the mixing of different chains (chromosomes), we also study the mixing within a single chain by calculating the contact probability, $P(s)$, as a function of the bead separation along the chain contour, $s$. We find that $P(s) \propto s^{-\gamma}$, where $\gamma$ is the exponent whose value indicates (when compared with the mixing analysis of the simulations) whether subdomains within a single chain mix or not. It has been reported that the contact probability scaling exponent ($\gamma$) for an unconfined phantom chain is $\gamma = 1.5$ and for a self-avoiding chain is $\gamma = 2.1$ [39, 40]. For a collapsed chain, a scaling exponent of $\gamma \approx 0$ indicates an equilibrium globule with a high degree of mixing of distant segments along the chain, while $\gamma = 1$ indicates a fractal globule characterized by very little mixing of distant segments [26, 41, 42]. An exponent of $\gamma = 0.5$ was deduced from the Hi-C experiments of a mitotic chromosome, with a computational polymer dynamics model suggesting intermediate mixing [42, 43]. Furthermore, $\gamma = 0.75$ observed from Hi-C contact maps of interphase chromosomes and compared with a computational polymer dynamics model suggests very slow mixing (glassy dynamics) [39, 44]. Also, recent studies suggest that chromosome X in mammalian cells forms a glassy globule with a scaling exponent of $\gamma = 0.72 \pm 0.2$ [45]. In this paper, we correlate the dynamics of chain mixing with the structural contact probability. The contact probability provides information about the mixing of different subdomains within each chain. Since the attraction within and among chains is the same, similar trends for the mixing of different chains are also observed. This is because if different regions of one chain are mixed due to attraction, then different chains will also mix in a similar manner. We have done this in the context of averaging the contact probabilities of four chains and have shown that the long-time steady-state value of $\gamma$ ranges between 0 (equilibrium mixing) and 1 (almost no mixing).

**Biophysical properties of chromatin at the mesoscale**: Chromatin (interphase chromosome) has been studied theoretically using models of polymers in both good and poor solvents [42, 46–48]. DNA is negatively charged and wraps around oppositely charged histone octamer proteins; however, there are also regions of non-histone-associated linker DNA, and chromatin still maintains a net negative charge [49]. Due to electrostatic repulsion of DNA and excluded volume, chromatin had often been considered as a self-avoiding or polymer in a good solvent. However, there are also many compelling reasons for treating chromatin (as opposed to DNA) as a self-attractive or polymer in poor solvents. The chromatin histone tails that are positively charged attract the negatively charged DNA linkers (in regions that can be far along the contour length of the chain, but close in three-dimensional space), leading to chromatin self-attraction and water acting as a poor solvent [49–54]. In addition, the HP1 chromatin-binding proteins are phase separated within the nucleus and may contribute to (hetero)chromatin condensation [55, 56]. Other studies have demonstrated the presence of a gel-like organization of chromatin that is conducive to self-attraction [57–62]. In this paper, we analyze the mixing dynamics of polymer chains (as models of chromosomes) by considering the polymer in both good and poor solvents by varying the strength ($\epsilon$) of a standard Lennard-Jones potential (see the Materials and methods section) that acts between the beads of the chain. If chromatin would behave as a polymer in a good solvent (random walk polymer, $R_g \sim N^{1/2}$ and self-avoiding polymer, $R_g \sim N^{3/5}$), its radius of gyration would typically exceed the diameter of the nucleus [63]; the nuclear envelope would then confine the polymer and prevent it from swelling. It is estimated that chromosomes take up 15 to 60% of the nucleus volume in *Drosophila* and 0.4 to 25% in humans [47, 64–67]. The different hydration of nuclei in different circumstances and organisms means that one must consider how chromosome volume fraction influences the dynamics of chromosome mixing. The polymers in our simulations (whether in good or poor solvent) are confined, and we study this effect by varying the volume fraction of chromosomes from $\phi = 0.001$ to 0.6. Within this range, the confinement diameter can be either smaller or larger than the radius of gyration of an equivalent random walk, allowing polymers to mix with or without constraint. In addition to the effects of hydration (chromatin volume fraction) and interactions, an individual chromosome is not a homogeneous chain (homopolymer). The chromosome is inhomogeneous even at the mesoscale since it contains both open (euchromatin) and compact (heterochromatin) domains [68]. Therefore, we have expanded the simulations of generic homopolymers to quantify how the heterogeneous nature of chromatin can prevent chromosomes from mixing. Our simulations of heteropolymers, where each chromosome is represented as a (non-periodic) block copolymer comprising both open and compact regions, reveal that chromosome mixing dynamics are slower in euchromatin regions and faster in heterochromatin regions.

Along with chromosomes, the nucleus also contains the nucleoplasm (mostly aqueous), the nuclear lamina, and chromosome-binding proteins. The nuclear lamina is composed of lamin proteins and lies on the inner surface of the nucleus (nuclear envelope). Its binding to the lamin-associated domains (LAD) of chromatin has been shown to slow down chromosome dynamics [69–71]. In this work, we also demonstrate through our simulations that the binding of LAD to the nuclear lamina results in increased chromosome-lamina contact, consequently reducing inter-chromosome contact. A recent experimental study conducted on live *Drosophila* indicates that, unlike fixed cells where chromosomes are uniformly distributed within the nucleus, chromosomes in live *Drosophila* tend to be organized near the lamina layer of the nuclear envelope [64]. Non-uniform, mesoscale distribution of chromatin had been previously discussed in the single-cell context [72, 73]. In the live fly experiments [64], overexpression of lamin C results in a shift from the peripheral to the central organization of chromosomes. These observations suggest that peripheral and central organizations are related to the

hydration of the live organism nuclei and that dehydrated nuclei (included by cell fixation and/or spreading on slides) can give only conventional organization in which chromosomes fill the nucleus. Theoretical analyses of the experimental work suggest that chromatin acts as a polymer in poor solvent (due to chromatin self-attraction) and that the various types of meso-scale organization are functions of hydration, chromosome-chromosome interactions, and chromosome-lamina interaction [46, 74].

## Materials and methods

In *Drosophila*, the genome is organized into four pairs of chromosomes. However, when simulating *Drosophila* genome, we only consider four chromosomes instead of 4 pairs because homologous chromosomes are paired at the molecular scale. To study the mixing of chromosomes in *Drosophila*, we simulate four coarse-grained, bead-spring polymer chains, each representing a single chromosome, with a total of $M = 8810$ beads per chain and $N = 35240$ beads in all four chains. Each bead represents 5 kbps of DNA and the associates histones, corresponding to a bead diameter of $\sigma = 30$ nm. The beads of each chain are connected to their nearest neighbors along the contour of the chain, by "springs" (harmonic potential) and the energy is written:

$$U_{\text{spring}} = \frac{1}{2} k_{\text{spring}} \sum_{\mu} \sum_{i} \left[ |\mathbf{r}_i^{\mu} - \mathbf{r}_{i+1}^{\mu}| - \sigma \right]^2,$$ (1)

where $r_i^{\mu}$ is the position of $i^{\text{th}}$ bead of the $\mu^{\text{th}}$ chain with $\mu = 1 \ldots 4$. $k_{\text{spring}}$ is the spring constant. Each chain has a persistence length of 5 beads, based on a previous estimate that interphase chromosomes consist of compact fibers with a diameter which is approximately 30 nm with a persistence length of 150 nm [65, 75–80]. We also simulated chains with a persistence length of 1 bead and compared the results with those whose persistence length was 5 beads. Since the bending energy determines the persistence length [81], this estimate allows us to write the bending energy of chromosome as:

$$U_{\text{bend}} = k_{\text{bend}} \sum_{\mu} \sum_{i} \left[ 1 - \cos \theta_i^{\mu} \right],$$ (2)

where $\theta_i$ is the angle between three adjacent beads within the chain. $k_{\text{bend}}$ is the bending stiffness of the chain which is related to the persistence length by $k_{\text{bend}} = \frac{l_p k_{\text{B}} T}{\sigma}$ [81]. Here $l_p$ is the persistence length, $k_{\text{B}}$ is Boltzmann constant, and $T$ is the absolute temperature. Apart from the nearest-neighbor spring interaction described above, any two beads (that can be anywhere along the contour of the chain) interact via the standard Lennard-Jones (LJ) potential ($U_{\text{LJ}}$) that depends on their three-dimensional spatial separation, with energy:

$$U_{\text{LJ}} = 4\epsilon \sum_{\mu \leq \nu} \sum_{i < j} \left[ \left( \frac{\sigma}{r_{ij}^{\mu\nu}} \right)^{12} - \left( \frac{\sigma}{r_{ij}^{\mu\nu}} \right)^6 \right],$$ (3)

where $r_{ij}^{\mu\nu} < 2.5\sigma$ and zero otherwise [46, 82]. Here, $r_{ij}^{\mu\nu} = |\mathbf{r}_i^{\mu} - \mathbf{r}_j^{\nu}|$ is the distance in 3D space between $i^{\text{th}}$ and $j^{\text{th}}$ beads of any chain and $\epsilon$ is strength of potential. We varied $\epsilon$ from 0 to 1 which can then account for no interactions ($\epsilon = 0$ for phantom chains), repulsive interactions (small values of $\epsilon$), and attractive interactions (larger values of $\epsilon$) between any two beads. The chains are confined by an impenetrable, spherical wall of radius $R_c$ that mimics the effect of the nuclear envelope. A confinement potential $U_{\text{nucleus}}$ is used to account for the hard-core repulsion between the beads and the spherical wall. We define the parameter $\phi$, as the volume fraction of chains within the sphere (corresponding to

chromosomes in the nucleus)

$$\phi = \frac{\text{Volume of chromosome chains}}{\text{Volume of confinement}} = \frac{N \times \frac{4}{3}\pi(\sigma/2)^3}{\frac{4}{3}\pi R_c^3},$$ (4)

$\phi$ ranges from 0.001 to 0.6, where small values of $\phi$ represent chains in large confinement volume and large values represent chains in small confinement volume. We performed simulations for $\phi$ = 0.001, 0.01, 0.1, 0.2, 0.3, 0.4, 0.5, and 0.6 which are respectively equivalent to confinement radii of $R_c$ = 164, 77, 36, 29, 25, 23, 21, and 20 in bead units ($\sigma$). The total energy of the system is given by the following equation:

$$U_{\text{tot}} = U_{\text{spring}} + U_{\text{bend}} + U_{\text{LJ}} + U_{\text{nucleus}}$$ (5)

To simulate the system, we used the molecular dynamics simulation package LAMMPS [83], in which Newton's equations are solved for particles in a solvent, represented by a motion in the presence of a viscous force and a Langevin thermostat to ensure an NVT ensemble. As we explained above, the short-time dynamical fluctuations of molecular motors are subsumed in effective temperature [17, 22, 23]. To obtain generic results, we first simulated chromosomal chains using the polymer model described above. Later, we modified the model to more realistically model the *Drosophila* genome. To do this, we included heterogeneity within each chain to represent the block copolymer structure of chromosomes with both euchromatin/heterochromatin, the presence of LAD/non-LAD chromosome domains, and attractive interactions between the LAD domains and the nuclear lamina (see S1 Text for details of our block copolymer model).

## Physical units of chromosome mixing models

In our coarse-grained model, each chromosomal bead represents 5 kbps of DNA, which represents 25 nucleosomes and their linker DNA. A nucleosome is modeled as a cylindrical structure with a diameter of 11 nm and a length of 5.5 nm [2]. The volume of 25 nucleosomes is comparable to the volume of a spherical bead of diameter to be 30 nm; therefore, we choose the diameter of chromosomal bead $\sigma$ = 30 nm. The time it takes such a bead to diffuse its own size ($\sigma$), is denoted as the Brownian time $\tau$, defined by $\tau = \frac{\sigma^2}{D_i}$, where $D_i = \text{k}_B T/\zeta$, is the diffusion coefficient of $i^{\text{th}}$ bead. The friction coefficient $\zeta$ is calculated by using the Stokes law for a spherical bead of diameter $\sigma$ in a solution of viscosity $\eta$ and given by $\zeta = 3\pi\eta\sigma$. When a bead's diameter $\sigma$ is known, the Brownian time is as $\tau = 3\pi\eta\sigma^3/\text{k}_B T$. Assuming that the viscosity of the nucleoplasm is similar to water, we find that with a viscosity $\eta$ = 1 cP [84], temperature $T$ = 310$K$, and a bead diameter of $\sigma$ = 30 nm, the Brownian time of a coarse-grained bead is $\tau$ = 60 μs.

## Initial conditions

During cell division, each chromosome condenses and separates from the other, and after cell division, two daughter cells are formed. This motivates our initial condition for which chains are initially condensed and separated in our simulation. For this biological reason and to obtain a lower bound on the mixing time, we used unknotted chromosome chains as the initial condition. Using moltemplate software [85], we created a single chain of 8810 beads and confirmed that it was unknotted (see the details in S1 Fig). We then placed four copies of this chain in a cubic box and compressed them into small spherical confinement using indented walls [83]. The Lennard-Jones interactions between the intra-chain were attractive, while the

inter-chain interactions were repulsive in the creation of the initial structure. This can be seen in S2 Fig, which illustrates the initially separated and condensed 4 chains in a confinement.

## Results

### Definition of the chromosome mixing index and its limiting values

The typical contact map of a genome shows intrachromosomal contacts along the diagonal and interchromosomal contacts along the non-diagonal (see Fig 1a). The Hi-C contact maps of the human and fly genomes show that non-diagonal contacts are relatively sparse while in the yeast genome, there are relatively more off-diagonal contacts [26, 86]. We analyze these genome contact maps to study how chromosomes mix and introduce a single, physical measure that we term the chromosome mixing index ($\alpha$), which quantifies in a coarse-grained

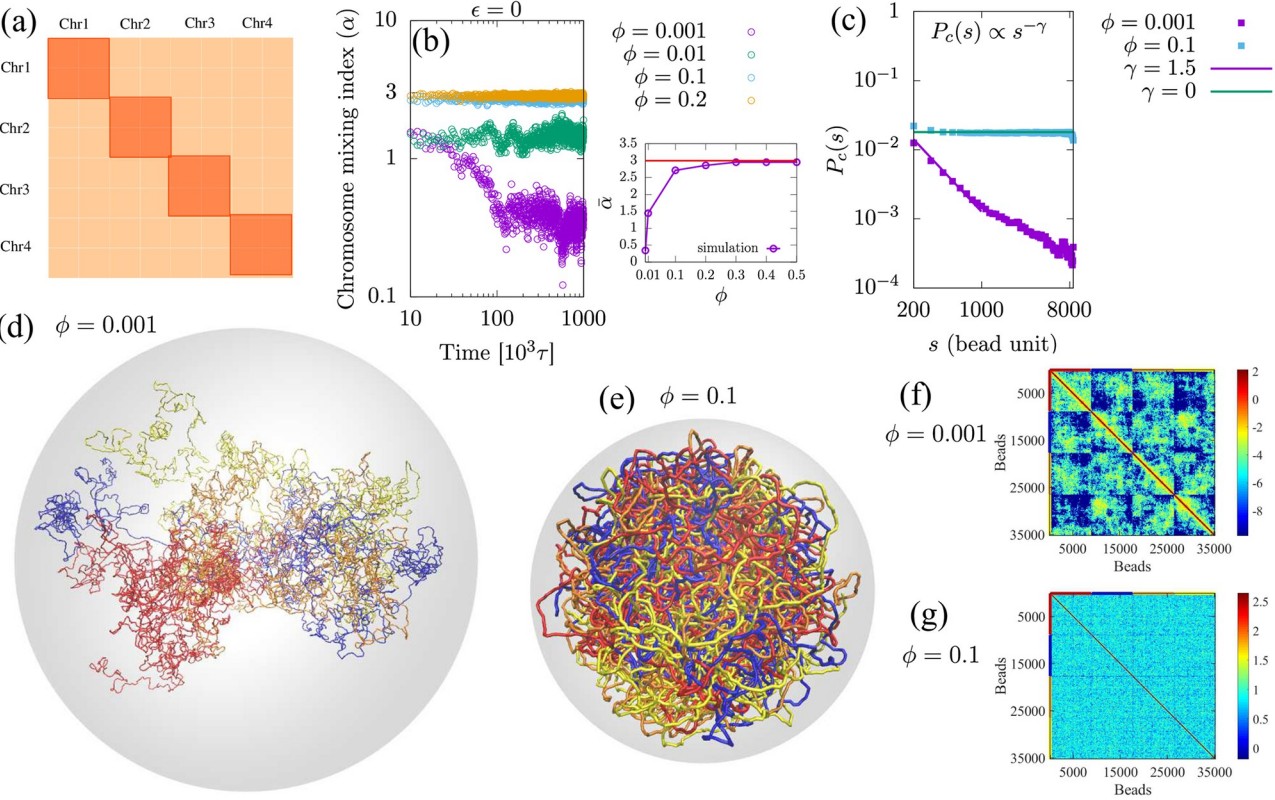

**Fig 1.** (a) A schematic diagram representing the contact map of a genome consisting of 4 chromosomes. Intra- and inter-chromosomal contacts are shown by dark and light orange colors, respectively. (b) The evolution of the chromosome mixing index, $\alpha$, over time, indicates that phantom chains quickly achieve equilibrium. The right panel of (b) is the time-average of the chromosome mixing index ($\bar{\alpha}$) as a function of the chain volume fraction ($\phi$) in the nucleus. For relatively large volume fractions $\alpha$ is approximately equal to three (the maximal value of $\alpha$ is 3 for this case of 4 chains) which indicates that phantom chains achieve maximal mixing in small confinement volumes. The chromosome mixing index is smaller than 3 for smaller volume fractions because the larger confinement volume (smaller volume fraction of chains), means that any two beads (even within the same chain) are less likely to be "in contact". (c) The average contact probability $P_c(s)$ within a single chain, averaged 4 phantom chains as a function of the bead separation distance $s$ with the scaling exponent $\gamma$ defined by $P_c(s) \propto s^{-\gamma}$. (d) Long time snapshot ($t = 10^6\tau$ time steps) of simulations of 4 phantom chains for a low volume fraction of chains $\phi = 0.001$, shows that the different chains are not mixed, since they can diffuse away from each other. The snapshot was zoomed out because its actual size was too large to be shown without taking up an excessive amount of space. (e) Late time snapshot ($t = 10^6\tau$ time steps) of simulations of 4 phantom chains for a relatively large volume fraction $\phi = 0.1$, shows that the chains are mixed. Note that in (d) and (e), each color represents a different chain corresponding to a different chromosome. The confinement volume in (d) and (e) is shown in light grey. (f) and (g) Contact maps calculated from phantom chain simulations for volume fraction $\phi = 0.001$ and $\phi = 0.1$ respectively. Note that contact maps were generated by applying the $\log_2$ transformation to the raw counts of the contact matrix and displaying the color bar in $\log_2$ ratios. In contact maps (f) and (g), vertical and horizontal colored lines have been added to distinguish the chains, which correspond to the colors of the chromosomes.

manner, the extent of chromosome mixing. In the Supporting Information (SI) S2 Text, we theoretically show that in the ideal limit (when there are no interactions), the chromosome mixing index is equal to $n - 1$, where $n$ is the number of chromosomes within the nucleus. For example, the maximum value of the chromosome mixing index for a genome with four chromosomes of the same size is 3. In general, $\alpha < n - 1$; the value of chromosome mixing indicates the actual mixing of the chromosomes. In order to allow for the maximal amount of chain (chromosome) mixing from simulation, we simulated phantom chains (non-interacting chains, $\epsilon = 0$), which mix quickly as there is no crowding effect due to the lack of volume exclusion (see Fig 1b).

We varied the volume fraction, $\phi$, of chains (representing chromosomes) within the nucleus and calculated the time-averaged chain mixing index, ($\bar{\alpha}$), from the last 300 frames out of a total of 1000 frames, as shown in the right panel of Fig 1b. The figure shows that phantom chains mix rapidly and attain a maximal value of the chromosome mixing index, ($\bar{\alpha} \approx 3$), for a system with four chains confined to a relatively small volume (a large volume fraction). For larger nuclear volumes (smaller volume fractions of chains), the chromosome mixing index is less than 3 because each chromosome freely diffuses across a large portion of the nucleus and is unlikely to be mixed. In S3 Fig, the ratio of the confinement size (diameter, $2R_c$) to the radius of gyration ($R_g$) of the phantom chain is plotted for various values of $\phi$. The results indicate that when $\phi \leq 0.01$, the confinement volume is relatively large, as $2R_c/R_g > 1$. Conversely, when $\phi \geq 0.1$, the confinement volume is relatively small, as $2R_c/R_g < 1$, and the chromosome mixing index for phantom chains reaches its maximum for these values of $\phi$. Note that for phantom chains, we compare confinement diameter with the radius of gyration corresponding to the length of a single phantom chain rather than the total length of all the chains since the various phantom chains do not interact.

We have calculated the contact probability from our phantom chain simulations in order to understand how the subdomains of each chain interact (see Fig 1c). Since we have 4 chains in our simulation, we calculated the contact probabilities of each chain and then took the average, so the final plot shows the average contact probability of 4 chains (see details in S3 Text). We fit the contact probabilities, $P_c$ as a function of the distance along the contour length of two beads (that belong to the same chain), $s$, with a power law: $P_c(s) \sim s^{-\gamma}$. We then found the scaling exponent $\gamma$ for the range $200 \leq s \leq 1000$ in bead units which is equivalent to the range of $1 \leq s \leq 5$ in mega basepairs. We have also calculated scaling exponent $\gamma$ for the range $1 \leq s \leq 8000$, which shows two exponents (at early and late times) when the confinement is small, and one exponent when the confinement is large (see S4 Fig). For $\phi = 0.1$, the contact probability is constant for large values of $s$, representing beads that are near the confinement surface. For smaller values of $s$, we obtain a scaling exponent of $\gamma = 1.5$ (see S4 Fig). This is consistent with an equilibrium globule structure [42, 43]. However, for $\phi = 0.001$, the contact probability of two beads separated by a contour length $s$ does not significantly saturate and is characterized by an exponent $\gamma = 1.5$, which is equal to that of an unconfined phantom chain where the Gaussian statistics (and our simulations presented in S5d Fig) result in a scaling exponent of $\gamma = 3/2$ [87]. This result also shows that the scaling exponent, $\gamma$, (for monomers far apart on the contour of the same chain) of phantom chains decreases with increasing chromosome volume fraction, $\phi$. The snapshots of the simulation in Fig 1d and 1e show that the phantom chains do not mix for larger confinement radii (smaller volume fraction $\phi = 0.001$) but are completely mixed for smaller confinement radii (larger volume fraction $\phi = 0.1$). Fig 1f and 1g are contact maps calculated for cases (d) and (e) respectively, obtained by counting beads whose centers are closer than $1.5\sigma$, where $\sigma$ is the bead diameter; for typical chromosomal bead, this corresponds to chromosomal beads closer than 45 nm being defined as "in contact". The contact maps are obtained by averaging over the last 500 snapshots (see details in S3 Text). The contact

maps calculated from the simulation of the phantom chains in Fig 1f and 1g show a heterogeneous probability distribution for larger confinement and a more homogeneous probability distribution for smaller confinement. From the results, it is clear that the phantom chains mix rapidly in smaller confinement (when $R_g > 2R_c$) and reach the maximal value of chromosome mixing index ($\alpha \approx 3$ for 4 chains).

In the following results, we consider interactions (both excluded volume and attractions) among the beads and calculate the chromosome mixing index by changing the chromosome-chromosome attraction strength $\epsilon$ and the chromosome volume fraction $\phi$ in our simulations (see the model for the definition of $\phi$ and $\epsilon$ in Materials and methods section).

## Effect of chromosome-chromosome attraction on chromosome mixing dynamics

After understanding the results of our simulations of the mixing of phantom chains, we next investigated the effects of the interactions (more realistic) between the beads and their effect on the mixing of different chains, while minimizing the effect of confinement volume. We thus performed simulations in a relatively large confinement volume—larger than the radius of gyration of an individual chain. We considered the situation where $2R_c/R_g > 1$, where $R_c$ is the radius of confinement and $R_g$ is the radius of gyration of a random-walk chain of $4M$ beads (with $M$ being the total number of beads in one chain) that is not confined (see S3b Fig). We simulated chains whose volume fraction in the confinement sphere was $\phi = 0.001$ and varied the strength of chromosome-chromosome interaction, $\epsilon$ (see the model in Materials and methods section and simulation parameters in S1 Table. To better understand how the attraction strength influences the condensation of a single-chain, we first simulated a single chain without confinement and calculated the radius of gyration as a function of the attraction of the beads (see S5 Fig). In the S6 Fig, we also discuss the analytical virial coefficient [88] for the LJ potential which we compare with the value obtained from the simulations at which the chains collapse due to the attractive interactions between the beads. For a given value of $\epsilon$, computation from the simulations of the radius of gyration of the chain indicates the range of attraction strengths for which the chain is open and those where the chain is collapsed. For $\epsilon < 0.4\,k_BT$, an isolated chain is open while for $\epsilon \geq 0.4\,k_BT$, an isolated chain is collapsed. $\epsilon_c = 0.4\,k_BT$ is the critical attraction strength at which the chain collapses. Note that for chains with a persistence length of one bead, the critical attraction strength was $\epsilon_c = 0.3\,k_BT$ (see S5b and S5c Fig) which is very close to the value analytically predicted by the vanishing of the second virial coefficient for the LJ potential (see S6 Fig). We also calculated the mean square displacement (MSD) as a function of time for different $\epsilon$ values to understand the dynamics of attractively interacting beads (see S5e and S5f Fig). For weak attraction strengths (still an open and not collapsed chain), $\epsilon = 0$ and 0.25, fits of the simulations are close to scaling of MSD $\sim \tau^{1/2}$. A scaling exponent of 1/2 applies to a Rouse chain [15] in which interactions are neglected. The fact that our simulation for finite $\epsilon$ gives an exponent smaller than 1/2 indicates the role of attractions [16]. However, for large attraction strengths, $\epsilon = 0.5$, 0.75 and 1, the simulations can be fit with values close to the scaling law MSD $\sim \tau^{1/4}$; the value of $\frac{1}{4}$ is indeed the MSD scaling exponent appropriate to polymer melts (reptation [12]). This is reasonable, since for large attraction strengths, the chains are very condensed and close to a melt state where the dynamics are slow since they involve reptation of the chain end and not diffusion of the chain in directions perpendicular to its local tangent (see S2 Table for variation in the value of scaling exponent of MSD as a function of volume fraction and interaction). High-throughput chromatin motion tracking experiments in living yeast cells have revealed that the mean squared displacement (MSD) of chromosomes follows a scaling law of MSD $\sim t^{1/2}$, indicating that they obey random

walk statistics [89, 90]. However, the MSD can be influenced by the organization and confinement of chromosomes and can vary among cells and cell types [16]. The scaling law for MSD of a polymer is $\text{MSD}(t) \sim t^{2\nu/(2\nu+1)}$, where $\nu = 1/2$ for a random walk chain and $\nu = 1/3$ for a collapsed, self-attractive chain, a chain in a poor solvent, or a chain in a small confinement volume. Our study (shown in S5 Fig) demonstrates how chromatin-chromatin interactions affect MSD. In S7 Fig, we calculated the average mean-squared displacement (MSD) of four chains over time for different attraction strengths, $\epsilon$. The range of mixing exponents $\beta$ is consistent with Rouse dynamics for weak interactions and a smaller exponent for stronger interactions [16, 91, 92].

To analyze the role of interactions in mixing, we simulated four chains as a function of $\phi$, starting with the initial condition for which all the chains are separated (see the Materials and methods section for the biological significance of this condition). As the simulation proceeds, the chains begin to mix so that intra-chain contacts decrease and inter-chain contacts increase with time. This reflects the non-equilibrium nature of chromosome mixing dynamics. In Fig 2a, we show snapshots of the simulations for different values of the attraction strength $\epsilon$ (see S8 Fig for snapshots of the case where the bead persistence length is unity, $l_p = 1$ bead). When the attraction strength of the chains is less than or equal to the critical strength for collapse ($\epsilon \leq \epsilon_c$), the chains are open and do not mix in a large confinement volume, because two beads (even in the same chain) are unlikely to be "in contact". At larger attraction strengths ($\epsilon = 1$ $k_B T$), each chain collapses but rarely mixes with the other because the kinetics slow down significantly as the time required for two interacting beads to detach is $e^{(z\epsilon/\,k_B T)}$, where $z$ is the number of nearest neighbors (not counting the two nearest neighbors to which each bead is bonded). In addition, the concentration of the chain is relatively larger in the collapsed state, which may make the chain kinetics more "jammed" for the condensed states at large values of the attraction. Fig 2b for $\epsilon = 1$ shows two mixing exponents, $\beta_1$ and $\beta_2$, indicating that the beads begin to mix at early times with a mixing exponent of $\beta_1 = 0.14$, while at late times, the mixing exponent is very close to zero ($\beta_2 = 0.04$), indicating a "jammed" situation. In summary, when polymer chains are in a large confinement volume (corresponding to small volume fractions $\phi \leq 0.001$), they are able to mix effectively only if they have moderate attractive interactions ($0.4 \leq \epsilon \leq 0.75$). This is demonstrated by the increase in the power law exponent that characterizes the chromosome mixing index, $\alpha \propto t^\beta$ as shown in Fig 2b. The right panel of Fig 2b also shows that the exponent value increases and then decreases as the attraction strength is increased. In S9 Fig, we have shown that the mixing index in disconnected beads with excluded volume interactions also follows a power law with time, but reaches its maximum value more quickly than for beads connected into chains. This suggests that the connectivity leads to slow mixing. Additionally, we have found that mixing is also slow (small mixing exponent) for disconnected beads that are strongly attractive, as seen in S10 Fig, highlighting the effect of attraction on mixing.

We calculated the contact maps by averaging the last 500 frames for different attraction strengths ($\epsilon$). This allows us to present a time-averaged view of the regions of high or low chromosome mixing (see Fig 2c). The contact probability ($P_c(s)$) calculated from the contact map is the normalized frequency of contact between beads spaced within the same chain, where $s$ is their separation along the contour length. We see that the probability obeys power-law scaling with $P_c(s) \propto s^{-\gamma}$, where $\gamma$ is the exponent (see Fig 2d). By calculating the contact probability from simulation results, we can find the contact probability scaling exponent for both an open and a collapsed chain. We first calculated the contact probability scaling exponent of a single, unconfined chain (for persistence lengths of both 1 and 5 beads) for various values of the LJ attraction strength, $\epsilon$ (see S4 Text and S5d Fig). The results show that for open chains ($\epsilon \leq \epsilon_c$) the contact probability scaling exponent is $\gamma > 1$ while for collapsed chains the contact

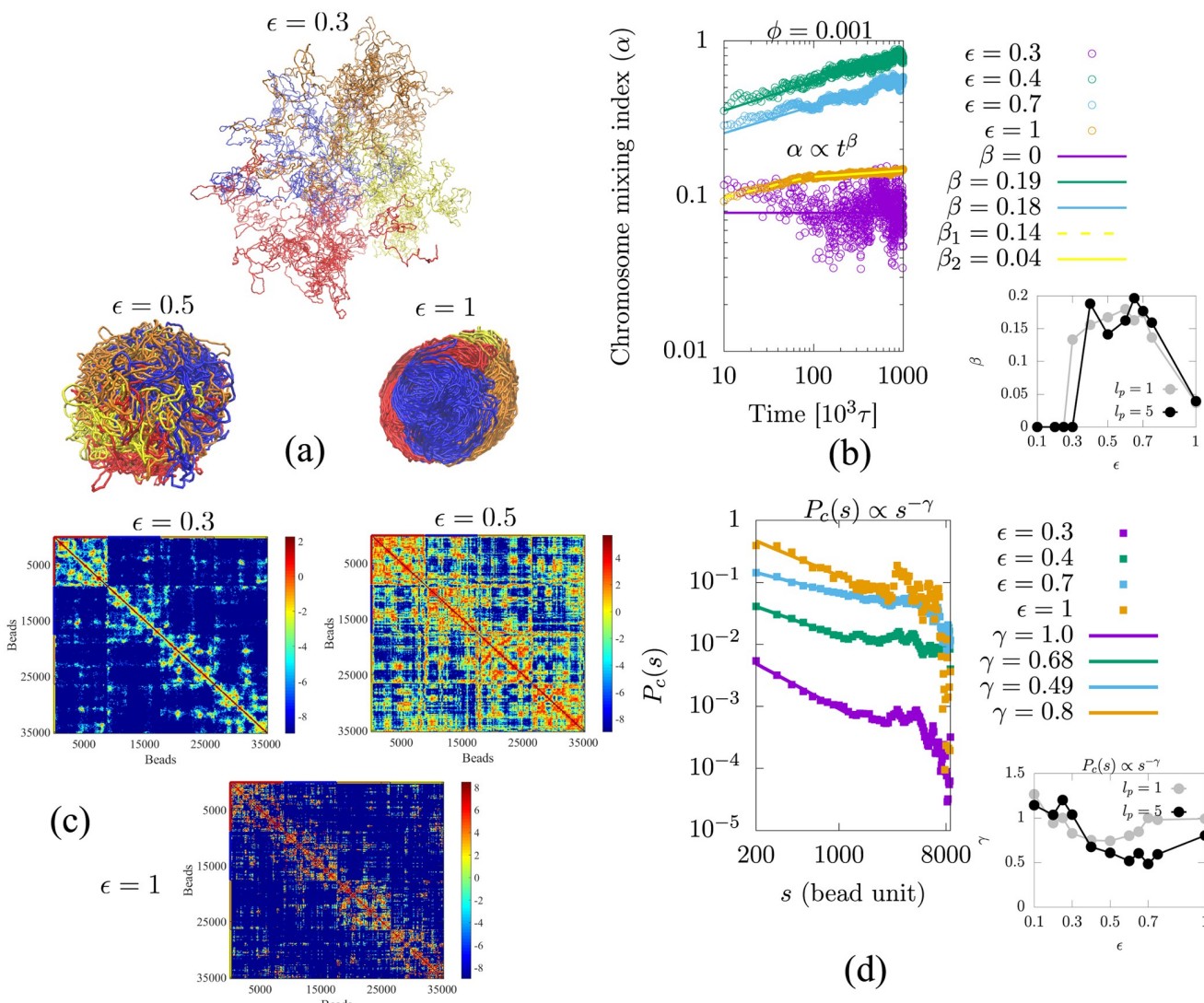

**Fig 2.** (a) Simulation snapshots at long times ($t = 10^6 \tau$) are shown for different attraction strengths ($\epsilon$) and for a volume fraction of $\phi = 0.001$. Each color represents a different chain corresponding to a different chromosome. For $\epsilon = 0.3$, the chains are open and not mixed. For $\epsilon = 0.5$, the chains are collapsed and mixed slowly. For $\epsilon = 1$, each chain is collapsed but rarely mixes with the others. Note that spherical confinement has not been shown in snapshots because it was too large relative to the chain size to be represented. (b) For small attraction strength ($\epsilon \leq 0.3$), chromosomes do not mix as they diffuse in the large confinement volume, with a mixing exponent of $\beta \approx 0$. When the attraction strength is moderate ($0.4 \leq \epsilon \leq 0.75$), the chromosome mixing index increases as a power-law with time, with the exponent $\beta$ characterizing the time exponent of the mixing. For strong attraction strength ($\epsilon = 1$), two mixing exponents are observed. Initially, chromosomes mix with an exponent of $\beta_1 = 0.14$, but later mixing slows down similar to "jamming", as indicated by the small value of the second exponent ($\beta_2 = 0.04$). From the right panel of (b), we find that the value of the exponent first increases and then decreases as the attraction increases. (c) Contact maps from the simulations calculated by averaging the last 500 frames are shown for different attraction strengths ($\epsilon$). In all these simulations, the volume fraction of chains is $\phi = 0.001$. (d) The average contact probability ($P_c(s)$) calculated within each chain is the normalized frequency of contact between beads at a separation distance of $s$ within a single chain. The contact probability shows a power-law relation $P_c(s) \propto s^{-\gamma}$, where $\gamma$ is the exponent. The right panel of (d) shows the scaling exponent ($\gamma$) for the contact probability within a single chain as a function of self-attraction strength, $\epsilon$ for persistence lengths $l_p = 1$ (gray color) and $l_p = 5$ (black color).

probability scaling exponent is $\gamma \leq 1$. Furthermore, for the collapsed chains, the scaling exponent $0 < \gamma < 1$ shows an intermediate level degree mixing that is a non-equilibrium state between mixing and no mixing. In order to analyze the contact probability behavior for the four chromosomes, we calculated the contact probability from each chromosome and plotted the average in Fig 2d. In the right panel of Fig 2d, we calculated the scaling exponent ($\gamma$) as a

function of attraction strength ($\epsilon$), which shows similar trends to that of a single chromosome, where $\gamma > 1$ for open chains and $\gamma \leq 1$ for collapsed chains. Collapsed chains show an intermediate level of mixing for $0 < \gamma < 1$. We also show simulation results of the mixing of four chains in relatively small confinement volume (see S11 Fig for simulation snapshots, mixing index, and contact probability). The results indicate that when confinement volume is small, the chromosome mixing index follows a power law for any attraction strength less than 1 ($\epsilon < 1$).

## Effect of chain volume fraction on mixing dynamics

To examine the effect of spherical confinement (chain volume fraction), we considered open chains (where the attraction is too small to cause collapse) and simulated weakly attractive chains ($\epsilon = 0.25$ k$_B$T) as a function of $\phi$. In Fig 3a, we present snapshots of simulations for different volume fractions of chains. For $\phi = 0.001$, corresponding to a large confinement volume with a relatively weak attraction strength ($\epsilon = 0.25$), the chains are separated. For $\phi = 0.2$, the chromosomes begin to mix until each chromosome approximately occupies the entire confinement volume. For $\phi = 0.6$, the smaller volume restricts the motions of the highly confined (and thus condensed) chains. The nematic ordering observed at this high volume fraction is related to the comparable size of the confinement radius and the persistence length of 5 beads for this simulation. In the S12 Fig we discuss this further and show the structure for a persistence length of one bead where separation, but almost no nematic ordering, is observed. In Fig 3b, the chromosome mixing index as a function of time shows a power-law increase with the exponent $\beta$ ($\alpha \propto t^\beta$) for various volume fractions of chains. For very low volume fraction $\phi \leq 0.001$ (or larger confinement volume, $2R_c/R_g > 1$) and weak attraction ($\epsilon \leq \epsilon_c$), the mixing exponent is $\beta \approx 0$, indicating diffusive chain behavior (as discussed in the previous section). For moderate volume fractions ($0.01 \leq \phi \leq 0.5$), the mixing exponent is $\beta > 0.12$, which indicates that the chains are mixing over time. For high volume fractions ($\phi \geq 0.6$), there are two mixing exponents: $\beta_1 = 0.19$ for short mixing times and $\beta_2 = 0.04$ for a longer time; this exponent indicates restricted motion of the chains due to the very small confinement volume. The simulation results give values of $\beta$ that range from 0 to 0.25, depending on the volume fraction (confinement). In the right panel of Fig 3b, the scaling exponent of the mixing index is plotted as a function of chain volume fraction, which shows that the exponent initially increases and then decreases with chain volume fraction. These power laws suggest that the mixing dynamics are very slow for very large and very small volume fractions (small and large confinement volumes, respectively). In Fig 3c, contact maps obtained from the simulations are calculated by averaging the last 500 frames and are shown for the different volume fractions, $\phi$. For $\phi = 0.001$, the contact map shows fewer intra- and inter-chain contacts which implies that the chromosome chains are relatively open and are hardly mixed. For $\phi = 0.2$, the contact map shows relatively more intra- and inter-chain contacts which implies that the chromosome chains are both relatively collapsed and mixed. For $\phi = 0.6$, the contact map shows relatively fewer intra-chain contacts which implies that the dynamics required to cause mixing is impeded; there are also fewer inter-chain contacts which implies relatively little mixing. In Fig 3d, we have plotted the mean contact probability as a function of the bead separation distance, $s$, within a single chain, calculated for the situation of 4 confined chains as the chain volume fraction ($\phi$) is varied. We find a power law behavior, with $P_c(s) \propto s^{-\gamma}$ with $\gamma$ exponent. In a large confinement volume (or small chain volume fraction $\phi = 0.001$), all the chains are open and the contact probability scaling exponent is $\gamma > 1$. Chains collapse for relatively small confinement (or high chain volume fraction $\phi \geq 0.01$), and the scaling exponent is $\gamma < 1$.

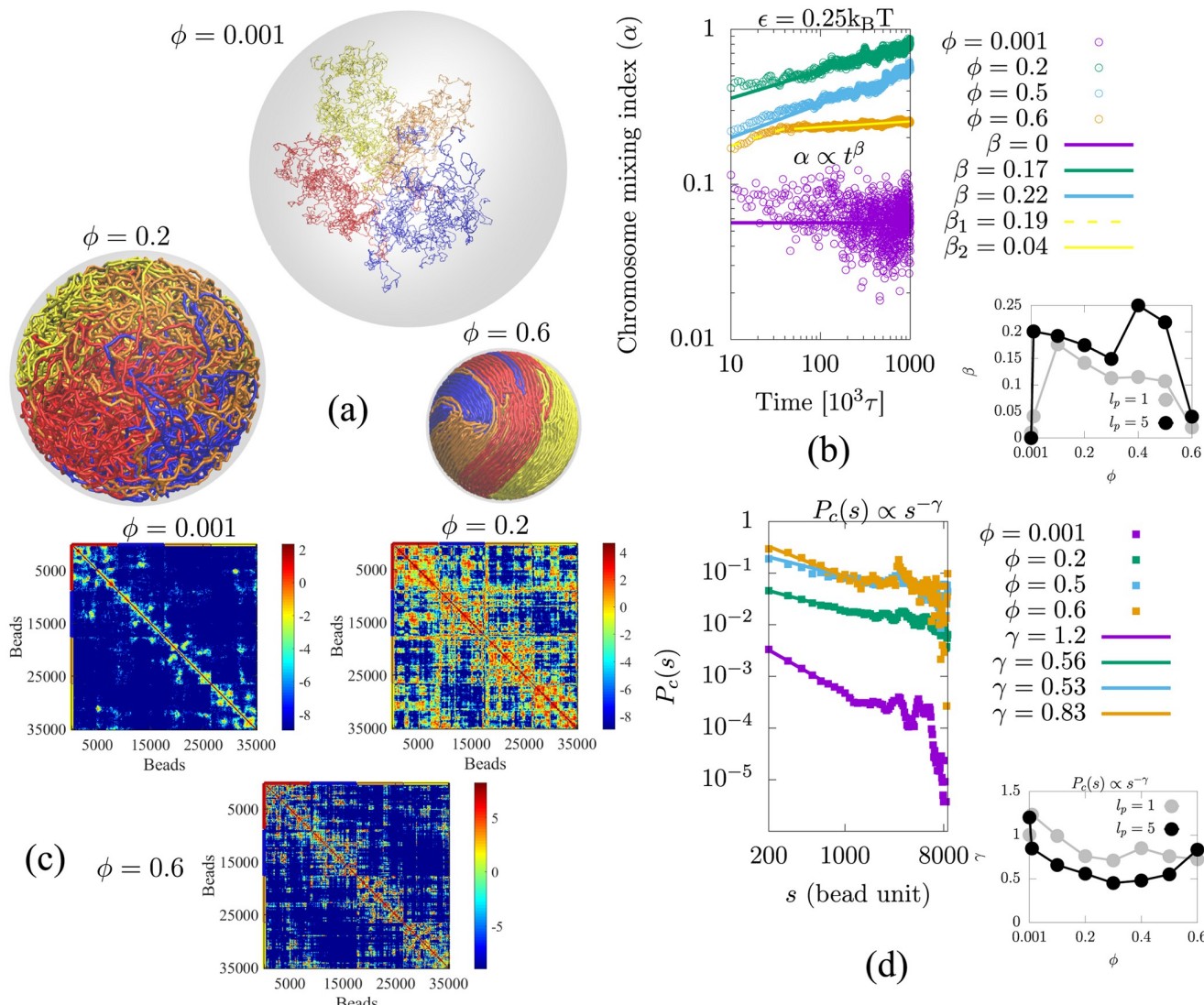

**Fig 3.** (a) Simulation snapshots are shown for different volume fractions of chromosome, $\phi$. For $\phi = 0.001$, corresponding to large confinement volume (shown in grey) with a weak attraction strength ($\epsilon = 0.25$), (which is below the value where collapse is observed) the chains are separated. For $\phi = 0.2$, the chromosomes begin to mix until each chromosome approximately occupies the entire confinement volume. For $\phi = 0.6$, the smaller volume restricts the motions of the highly condensed chains. (b) The mixing index, represented by $\alpha$, increases over time according to a power law, with the exponent $\beta$. This holds true for moderate volume fractions ($0.01 \leq \phi \leq 0.5$). However, at very small volume fractions ($\phi = 0.001$), the mixing exponent is $\beta \approx 0$, indicating diffusion of chains. At larger volume fractions ($\phi = 0.6$), the motion of chains is restricted due to small confinement volume, resulting in two exponents: $\beta_1$ and $\beta_2$. Right panel of (b) The exponent first increases and then decreases as the chain volume fraction is increased. These power laws suggest that the mixing dynamics is slow for very large and very small volume fractions (small and large confinement volumes, respectively). (c) Contact maps obtained from the simulations, calculated by averaging the last 500 frames, are shown for the different volume fractions, $\phi$. (d) The single-chain contact probability ($P_c(s)$) is plotted with different colored points for different volume fractions of the chains, $\phi$. The line corresponding to each color indicates a power-law relationship. The right panel of (d) is the scaling exponent $\gamma$ as a function of chromosome volume fraction $\phi$ for persistence lengths $l_p = 1$ (gray color) and $l_p = 5$ (black color).

## Extrapolated time for full mixing of chains

We now simulated chain mixing as a function of both the volume fraction (confinement) and the attraction strength. The results of simulations for each pair of volume fractions (confinement) and chain-chain attraction strengths ($\phi$, $\epsilon$) are shown for $t = 10^6\tau$ time-steps where $\tau = 60$ μs; this corresponds to about 60 seconds of real-time (see the Materials and methods

section). The chromosome mixing index did not reach its maximal value ($\alpha = 3$ for 4 chains) for any pair of $(\phi, \epsilon)$, which means that the chains did not completely mix during this time interval. From these simulations, we used an extrapolation of the power-laws for the mixing index as a function of time to estimate when the four chains will be completely mixed. From the relation $\alpha \propto t^{\beta}$, we extrapolated the time (t) for which $\alpha(t) \approx \alpha_{max} = 3$ where $\alpha_{max}$ is the ideal limit of $\alpha$ for which chains are fully mixed. In the heat map shown in Fig 4a, we present the extrapolated time for the chains to fully mix for different pairs of $(\phi, \epsilon)$. The heat map displays the time in seven colors according to VIBGYOR (Violet, Indigo, Blue, Green, Yellow, Orange, Red) colors pattern for the various pairs of $(\phi, \epsilon)$; this shows which pairs show complete mixing and on which time scale. When the attraction is weak ($\epsilon \leq \epsilon_c$), the chains do not reach equilibrium if the volume fraction is low ($\phi \leq 0.001$) or ($2R_c/R_g > 1$ in S3b Fig). This is because the chains diffuse in the large confinement volume and never fully mix. The extrapolation time goes to infinity since the mixing exponent $\beta \approx 0$. These regions are represented by red colors (lower left regions) in the heatmap, as any mixing that takes longer than $t = 10^{16}\tau$ is considered red. Interestingly, the value $\epsilon = 0.3$ corresponds to the attraction at which the second virial coefficient of the LJ potential goes to zero (see S6 Fig). From our calculations, we see that the pair $\phi = 0.4$ and $\epsilon = 0.3$ shows the fastest mixing time $t = 6.2 \times 10^7 \tau$, which corresponds to 1 hour in real time (greater than typical cell cycle time in early *Drosophila* embryogenesis but lower than the typical cell cycle in late embryos [18]).

These calculations suggest that the dynamics of chromosome mixing is a slow process and that most cells will likely divide before chromosomes are completely mixed. Of course, partial mixing does occur, as discussed above and in [5, 29]. Different orders of magnitudes of the mixing time correspond to the different colors (from violet to red) in the heat map. The violet to green colors in the heat map corresponds to the range of mixing times of the order of $10^7$ to $10^{12}\tau$ for different pairs of $(\phi, \epsilon)$. Since yellow and orange do not appear in the heat maps, this

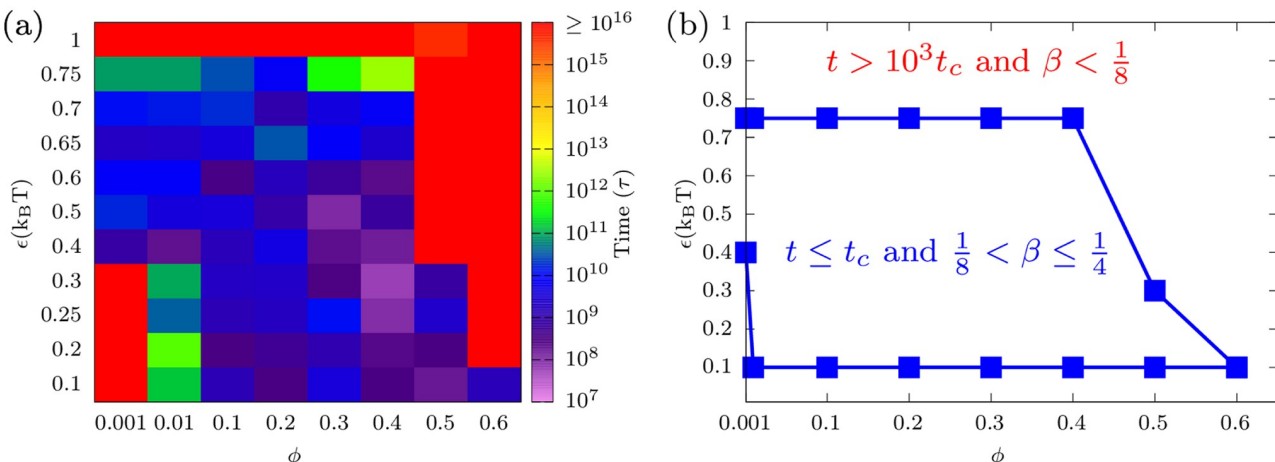

**Fig 4.** (a) Heat map of mixing time for different pairs (88 pairs) of volume fractions (confinement) and self-attraction ($\phi, \epsilon$) where $0.001 \leq \phi \leq 0.6$ and $0.1 \leq \epsilon \leq 1$. The different colors indicate the order of the time when the four chains in our model are approximately maximally mixed ($\alpha \approx 3$). Various colors represent different times when chromosomes completely mix. The time difference between the mixing order that is shortest (represented by the green color) and longest (represented by the red color) is about 1000. The extremes of the green color region, for which the pair $\phi = 0.4$ and $\epsilon = 0.75$ completely mix in time $t = 2.68 \times 10^{12}\tau$, corresponds to 5 years and is denoted as the critical time. Note that the red color signifies extremely slow mixing, three orders of magnitude in time longer than $t_c$, while the other colors show mixing times shorter than (due to overlap with no reptation) or on the order of (due to reptation) $t_c$. (b) State diagram showing the transition from slow mixing ($t \leq t_c$) to almost no mixing within $t_c$ ($t > t_c$) of the chains. The bars represent the simulation results, while the line serves as a guide for the eye. In the heat map (a), we observe violet, indigo, blue and green regions for which the chains are completely mixed by confinement and/or self-attractions for times shorter than the critical time ($t_c$), and the red region for which the mixing time (due to confinement and/or self-attractions) is 3 orders of magnitude longer than the critical time.

means that the chains do not mix within the times of the order of $10^{13}$ to $10^{14}\tau$ for any pair of $(\phi, \epsilon)$ values that were studied. In the heat map, the red color indicates the largest order of magnitude of time ($t \geq 10^{15}\tau$) for complete mixing; for smaller times, points that are red in the heat map rarely mix. Based on the heat map colors, we can divide the time scales into two regimes: (1) slow mixing that eventually fully mixes in a long time $t_c$ (to be defined), and (2) rare mixing that never fully mixes in a long time, $t_c$. The time in which chains mix within a time less than $10^{12}\tau$ is taken to define the critical time ($t_c = 2.68 \times 10^{12}\tau$ which applies to the pair $\phi = 0.4$ and $\epsilon = 0.75$). We choose this value because corresponds to 5 years and is similar to previous work where the reptation time for *Drosophila* genome is estimated as being 5 years [17]. There is a jump of three orders of magnitude in the mixing time for pairs of $(\phi, \epsilon)$ that have longer mixing times. In Fig 4b, a state diagram is shown, displaying the regions for pairs of $(\phi, \epsilon)$ in which chromosomes mix, albeit slowly ($t \leq t_c$), and for pairs where chromosomes mix rarely ($t > 10^3 t_c$). The state diagram also shows the scaling exponent $\beta$ of the chromosome mixing index. For $t \leq t_c$, the scaling exponent $\beta$ is in the range 1/8 to 1/4, and for $t > t_c$, $\beta$ is less than 1/8 (see S13 Fig for values of $\beta$ for different pairs of $(\phi, \epsilon)$). It is interesting to note that a bead of chain moves with a mean square displacement that scales $t^{1/2}$ for Rouse diffusion and with mean square displacement that scales $t^{1/4}$ in reptation motion [12, 13]. The root mean square (RMSD) displacements thus scale with the exponents 1/4 and 1/8 that correspond to the values obtained from the mixing index. These results suggest that the contact map dynamics (that show mixing) can be understood in terms of the single bead dynamics for the limits of non-interacting chains (Rouse dynamics with RMSD of 1/4) and strongly interacting and condensed chains (reptation dynamics with RMSD of 1/8). While the details of the mixing dynamics are complex, our observation of a power law for the increase in the mixing index suggests that there may be a dominating effect. We identify this effect with the power-law time dependence of the RMSD, represented by $\lambda/2$. We found, as shown in S2 Table, that the exponents $\lambda/2$ and $\alpha$ are very similar and change in response to the different polymer-solvent, and polymer volume fraction conditions, such as polymers in a good solvent, poor solvent (self-attractions), high concentration (reptation, similar to melts).

## Effects of the lamina on the mixing of *Drosophila* chromosomes

The generic model shown above studied the mixing of four polymers of the same size. Here, we model the genome of *Drosophila* in a more realistic manner, taking into account the sizes of its four chromosomes: Chr2, Chr3, Chr4, and ChrX, with sizes of 60.5, 68.8, 4.5, and 42.4 Mbps, respectively [93]. We model the *Drosophila* chromosomes as different-sized bead-spring polymers, with Chr2 comprising 12,100 beads, Chr3 comprising 13,760 beads, Chr4 comprising 900 beads, and ChrX comprising 8,480 beads. Each bead corresponds to 5 kbps of DNA (see Fig 5a). In the previous sections, we have shown that open polymers ($\epsilon \leq \epsilon_c$ below the collapse transition) mix slowly in relatively small confinement volumes ($\phi \geq 0.1$), while attractive and hence condensed polymers ($\epsilon = 0.5$) mix more readily in experimentally reasonable times. In *Drosophila*, each chromosome comprises both relatively open (euchromatin) and relatively condensed (heterochromatin) domains (see Fig 5a). To account for this situation, we extend our previous results for homopolymers to heteropolymers that comprise blocks of both open and condensed polymers and investigate how the heterogeneity affects the mixing of several chains in confinement. Each heterogeneous chromosome (blocks of euchromatin and heterochromatin) is modeled using two types of beads with different self-attractions corresponding to euchromatic and heterochromatic regions. The attraction between any two "euchromatic" beads is $\epsilon_{EE} = 0.25 \, k_B T$, while between any two "heterochromatic" beads, the interaction is larger, with $\epsilon_{HH} = 0.5 \, k_B T$. In this simulation, euchromatic and heterochromatic beads do not

attract and have only excluded volume interactions. In the simulation snapshot shown in Fig 5b, euchromatic regions of the different polymers mix less, while their heterochromatic regions mix to a greater extent. The contact map in Fig 5c, calculated from the block copolymer model, shows more than four regions along the diagonal (as in Fig 1a) due to the separate intra-chain mixing of the euchromatic and heterochromatic blocks, corresponding to the chromosomes in Fig 5a. From these results, it is clear that heterochromatin regions in the genome largely mix due to their strong self-attraction [20]. This serves to increase the overall mixing of the chromosomes compared to homopolymers with smaller attraction strengths.

Experimental studies have shown that specific chromosome regions, denoted as lamin-associated domains (LADs), distributed along the chromosome [69], are tethered to the nuclear lamina. To account for the role of the lamina in chromosome mixing, we included the bonding of some fraction of the beads (representing the LADs) to the nuclear lamina within our block copolymer model for *Drosophila*. The nuclear lamina is modeled as a thin layer of

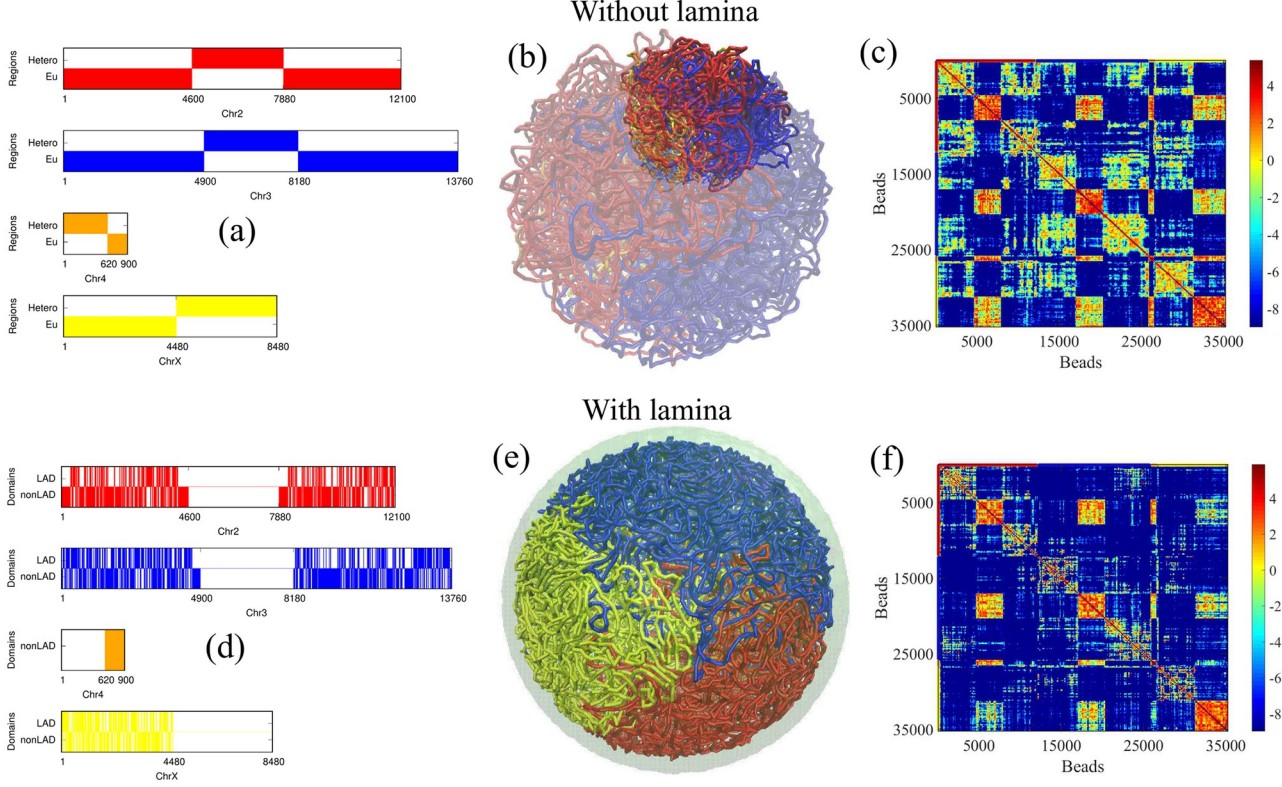

**Fig 5. Here, we present results from our block copolymer model, which simulates the chromosomes of the *Drosophila* genome.** We represent the chromosomes as chains with two types of beads, each with different attraction strengths corresponding to euchromatic (relatively weak attraction) and heterochromatic (relatively strong attraction) regions. (a) The distribution of euchromatic and heterochromatic regions along each chromosome is based on experimental data from [93]. (b) A simulation snapshot of the chains mixing, where the euchromatic beads are transparent, allowing only the heterochromatic beads to be clearly visible. We observe that heterochromatic regions of different chains mix due to their large self-attraction. (c) The contact map calculated from the block copolymer model shows more than four regions along the diagonal (as in Fig 1) due to the separate intra-chain mixing of the euchromatic and heterochromatic blocks corresponding to the distribution shown in (a). To understand the role of the binding of chromatin to the lamina in chromosome mixing, we modeled the bonding of certain chromosome segments (LADs) to the nuclear lamina using our block copolymer model for *Drosophila*. (d) The distribution of LAD and non-LAD domains along each chromosome is based on experimental data from [94]. (e) A simulation snapshot for chromosome volume fraction $\phi = 0.1$ is presented, showing that in the presence of lamin (green), the four chromosomes Chr2 (red), Chr3 (blue), Chr4 (orange), and ChrX (yellow) of the *Drosophila* chromosomes separate into distinct regions and do not mix. (f) The contact maps for case (e). Comparing this to the contact map for the same block copolymer model in the absence of binding to the lamina in Fig 5c, we observe a smaller probability of contact in the off-diagonal regions. From these results, it is clear that binding to the nuclear lamina decreases the mixing of chromosomes.

"lamin beads" localized near the surface of the spherical confinement. In Fig 5d, we have shown the distribution of LADs (lamin-associated domains) and non-LADs along each chromosome based on the experimental data [94]. Note that this data is available only for the euchromatic regions of the chromosomes. Besides LADs and non-LADs, each chromosomal bead corresponds to either euchromatin or heterochromatin according to our block copolymer model. Fig 5e shows a simulation snapshot for chromosome volume fraction $\phi = 0.1$. We see that the chains separate in distinct territories and rarely mix due to lamin-LAD attraction at the nuclear periphery. We have calculated the contact map in the presence of lamin and compared it to the contact map for the same block copolymer model in the absence of binding to the lamina in Fig 5f. In the presence of lamin binding, the simulations show less contact in the off-diagonal regions. In S7d Fig, we computed the mean-squared displacement (MSD) of chains (chromosomes) using our block copolymer model modified to include the attraction of chromatin to the nuclear envelope (as a model of the lamina). Our findings reveal that the exponent of MSD $\sim \tau^{0.26}$ is smaller compared to homopolymers that do not interact with the nuclear envelope (as a model of the lamina) (see S7 Fig). This result supports previous experimental research indicating that the loss of lamin A function increases chromatin dynamics in the nuclear interior [95]. From these results, it is clear that binding to the nuclear lamina reduces the mixing of chromosomes.

## Discussion

Our findings suggest that the chromosome mixing index, $\alpha(t)$, is useful as a single, time-dependent parameter that quantifies the slow (and sometimes, extremely slow) mixing of different chromosomes confined to the nucleus. For ideal chains within small confinement volumes, the mixing index of chromosomes is maximal; a given chromosome mixes with the $n - 1$ other chromosomes. The value of the chromosome mixing index depends on the volume fraction and chromosome interactions, which change the scaling law of how chains mix as a function of time. The chromosome mixing index increases slowly over time: $\alpha \sim t^{\beta}$ with an exponent of $\beta$. For slowly mixing chromosomes, our simulations predict a mixing exponent of $\beta = 1/8 - 1/4$ depending on $(\phi, \epsilon)$. Unconfined polymers collapse when their self-attraction strength is less than or equal to a critical strength ($\epsilon \leq \epsilon_c$) related to the vanishing of the second virial coefficient at the theta-point, at which the polymer first collapses (see S6 Fig). In contrast, confined polymers collapse when their confinement diameter is smaller than the radius of gyration of the unconfined polymer ($2R_c < R_g$). Recent studies have revealed the role of SMC (structural maintenance of chromosomes) proteins, such as condensin II, in the formation of chromosome territories [10, 96]. Research has shown that SMC proteins are responsible for controlling mixing via a mechanism called loop extrusion [97, 98]. When SMC is knocked down in *Drosophila*, there is an increase in chromosome mixing, suggesting that these proteins play a role in the spatial organization of chromosomes within the nucleus. A recent theoretical study has shed light on how SMC proteins form large loops in DNA [99]. However, the inclusion of such loop motors and their effects is outside the scope of our paper. Our simulations show that when the mixing exponent ($\beta$) is greater than 0.1 ($\beta > 0.1$), indicating weak attraction between chromosomes in small confinement volume, chromosome mixing is similar to what would be expected without condensin II. However, when $0 < \beta < 0.1$, indicating strong attraction between chromosomes, chromosome mixing is similar to what would be expected with condensin II present. We suggest that on long-time scales, the activity of condensin II may be captured by an effective self-attraction of the beads. Additionally, we found that lamin proteins, which are involved in chromosome organization and play a role in the formation of chromosome territories, can also reduce chromosome mixing.

The overlapping of partially mixed chains (which represent chromosomes) can be seen in our simulation results by focusing on two chains, Chr2 and Chr3, from the case shown in Fig 6a and counting the number of beads of Chr2 that lie within the overall contour of Chr3, and vice versa (see Fig 6b). The overlap of these chains at their edges indicates partial mixing [4, 5]. In Fig 6e, we calculated the chromosome mixing index from the Hi-C experiment data of the *Drosophila* genome [27]. The chromosome mixing index ($\alpha$) that we calculated from the experimental data is $\alpha = 0.3$, and its maximum value was calculated theoretically from a generalization of the equation in the S3 Text (to chromosomes of different sizes), considering heteropolymers, which is approximately 4 ($\alpha_{max} = 3.9$). We determined the chromosome mixing index ($\alpha$) using our block copolymer model for three different volume fractions ($\phi = 0.1, 0.2, 0.3$) in the presence of attractive interactions of the chromatin with the nuclear envelope (as a model of the lamina). The values of $\alpha$ obtained were 0.18, 0.24, and 0.34, respectively. Notably, our calculation of $\alpha$ for $\phi = 0.3$, as shown in Fig 6d, was in good agreement with the experimental value presented in Fig 6e. It is worth noting that recent experiments on *Drosophila* muscle nuclei have suggested a chromatin/nucleus volume ratio of 0.31 [64]. The simulated contact map we obtained in Fig 6d displays features that are similar to the Hi-C contact map in Fig 6e, as both maps show five distinct domains (squares) along the diagonal for chromosomes 2L, 2R, 3L, 3R, and X. This outcome was achieved even though we used identical intra/inter chain interaction strengths in our simulation and applied the same interaction strength

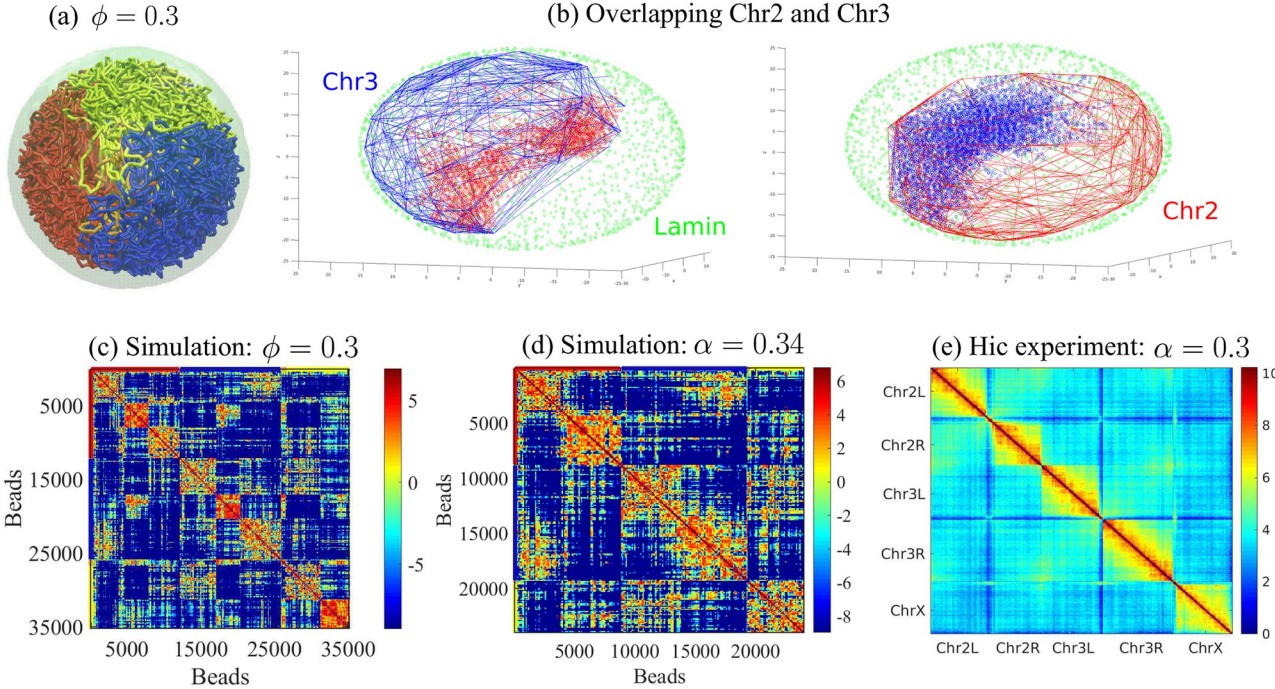

**Fig 6.** (a) Snapshot of simulation of block copolymer model in the presence of nuclear boundary (lamin) attractions for a chain volume fraction of $\phi = 0.3$. (b) Overlapping between Chr2 and Chr3 is shown. Lamin beads are shown by green points. Chr2 and Chr3 regions in 3D are represented by the line, and overlapping regions of Chr2 and Chr3 are shown by red and blue points. (c) The contact map for case (a). (d) The contact map and chromosome mixing index are calculated for the simulation shown in (a) by omitting the heterochromatin regions. Note that in the contact maps, vertical and horizontal colored lines are added to distinguish the chains, which correspond to the colors of the chromosomes. (e) Contact map and chromosome mixing index are calculated from the Hi-C experiments data of *Drosophila* genome [27]. Note that the contact map calculated from our simulation (in (d)) is noisy but similar to the experimental contact map (in (e)). Both maps show that the euchromatic parts of chromosomes 2L and 2R are separated with little mixing, as well as chromosomes 3L and 3R, as evidenced by the five red squares on the diagonal of (d) and (e). This is interesting because it was achieved using only one interaction parameter in the simulation (attraction strengths between all euchromatic beads were the same in our simulation).

for sub-chains Chr2L and Chr2R. We also calculated the Pearson correlation coefficient ($-1 \leq \rho \leq 1$) between the Hi-C and simulated contact maps, giving us a quantitative measure of how similar the two maps are [100]. A high correlation coefficient indicates a strong similarity between the two maps, while a low correlation coefficient indicates a weak similarity. The Pearson correlation coefficient between the simulated and experimental contact maps was 0.9. This indicates that the extension of the theoretical model to include different euchromatin/heterochromatin interactions and LAD/non-LAD interaction with the lamina yields mixing whose index is comparable to Hi-C experiment data. In S14 Fig, we calculated the chromosome mixing index ($\alpha$) using Hi-C data from measurements during the different stages of *Drosophila* development [101]. We found that chromosome mixing gradually increased from nuclear cycle 12 to nuclear cycle 14.

Although our calculated contact maps are noisy, it may be due to the use of a minimal number of parameters in our simulation in order to maintain conceptual simplicity. This is because scaling laws are often robust and not sensitive to many of the microscopic details. In our simulations, we always used the same interaction for intra-chromosome and inter-chromosome interactions. Previous studies that simulated genomes used data-driven models, which utilized Hi-C contact maps to establish contacts between intra-chromosomal and inter-chromosomal regions, with the aim of comparing them to their corresponding simulated contact maps [20, 30, 102]. In one simulation, we decreased inter-chromosome interactions compared to intra-chromosome interactions and were able to generate a contact map that more closely resembles Hi-C genome data, where each separated chromosome can be seen in a diagonal pattern (see S15 Fig). Also, we used simple LJ and harmonic potentials, which are commonly used in polymer simulations. Scaling laws depend on generic regimes (e.g., in classical polymer physics: dilute, semi-dilute, melt), and the power-law exponents are not sensitive to many of the microscopic details. We have also shown that the power-law behavior did not change when using FENE and shifted LJ potentials (see S16 Fig) [48]. Even though we began simulations with unknotted chains, chain mixing can result in knots forming between chains. Type II topoisomerase enzymes are known for their ability to remove DNA knots [38]. Our simulation results suggest the absence of knots in two important limiting cases: (i) Phantom chains (see Fig 1) that may correspond to chromosomes in the presence of cutting enzymes such as Type II topoisomerase [38]. The simulations show that the chains mix relatively quickly when confined, but once the confinement is removed, they remain un-mixed due to entropy and the absence of interactions between chains. This suggests a lack of knots. (ii) Strongly attractive chromatin [49, 52], where we show (see Fig 2) that the structure is characterized by a contact probability that scales with an exponent $\gamma \approx 1$, which is characteristic of a fractal globule. The fractal globule has relatively little mixing of different sections of the chain, again suggesting the absence of knots [87]. Our results in S5a Fig for the case of $\epsilon = 1$ and persistence lengths of 1 and 5 beads show that the subchains corresponding to different colors are separated.

We used the same dynamical simulations to calculate the scaling exponent of the contact probability, $P_c(s)$, for segments of a single chain separated by a contour length $s$: $P_c(s) \sim s^{-\gamma}$ with $\gamma \leq 1$ for collapse chains, independent of whether they collapse due to attraction strength ($\epsilon$) or due to small confinement volume (relatively large volume fraction, $\phi$). Open, weakly, or non-attractive polymers (as a model of euchromatin) mix even in relatively small confinements (such as the nucleus), while strongly attractive polymers that collapse and become highly condensed (as a model of heterochromatin) mix extremely slowly. The scaling exponent of the contact probability factor $\gamma$ is useful to understand the mixing of subdomains within single chains, whereas the scaling exponent of the chromosome mixing index $\beta$ is useful to understand the mixing of different chains. We note that it is possible that there are only a few universality classes for these scaling exponents. For example, the dynamical exponent may

have one value when Rouse dynamics is applicable (root mean squared displacement (RMSD) of a bead varies as $t^{1/4}$) and another for the strongly attractive regime when the chains are highly condensed and the dynamics occur via reptation (RMSD varies as $t^{1/8}$). Indeed, these are the values for the dynamical mixing exponent $\beta$ at the theta-point (vanishing of the second virial coefficient) and for strongly attractive chains, respectively (see S3 Table for a comparison between the scaling exponents of RMSD and mixing index). This agreement indicates how the polymer dynamics via the RMSD exponents for bead motion relates to the dynamical evolution of the entire contact map—an important link between polymer physics and experimental Hi-C biology. The variation of the exponent between these limits may be due to crossover effects, although we note predictions of the RMSD of interacting polymers that are neither Rouse-like nor collapsed give scaling exponents that vary with the classes of interactions [16].

## Supporting information

**S1 Text. Block copolymer model for *Drosophila* genome: Incorporating chromosome size, euchromatin and heterochromatin domains, and the interaction of lamin with LAD and non-LAD regions.**
(PDF)

**S2 Text. Chromosome mixing index.** This text describes the definition and formula of the chromosome mixing index.
(PDF)

**S3 Text. Calculating the contact map, contact probability, and chromosome mixing index from simulation data.**
(PDF)

**S4 Text. Unconfined single chain as a function of self-attraction strength.** In this text, for an unconfined single chain, we discuss how the attraction of the beads characterized by the parameter $\epsilon$ affects the radius of gyration ($R_g$), mean-square displacement (MSD) as a function of time, and the contact probability scaling exponent ($\gamma$).
(PDF)

**S1 Fig. Initial structure of a single chain and late-time snapshot of self-avoiding chain from simulation.** (a) Upper left: The initial structure of a single chain of 8810 beads was generated using the moltemplate software from python code interpolate_coords.py [85]. This code uses cubic spline interpolation to create a smooth polymer structure within a given size of the cubic box. Upper right: The initial structure was colored along its length using 10 different colors, ranging from blue at one end to red at the other. (b) A late-time snapshot of a self-avoiding chain from simulation. The simulation was conducted with a persistence length of 1 bead and a simple Lennard-Jones potential with a strength of $\epsilon = 1 k_B T$ and a cutoff of $r_c = 2^{1/6}\sigma$. The snapshot shows an open chain structure with separated colors, indicating that the initial structure was unknotted.
(PDF)

**S2 Fig. Initial structure of four separated chains in a small space.** The initial structure consists of four separate chains confined in a small space. By using moltemplate [85], one chain was generated, and then, by using packmol [103], four copies of chains were placed in a cubic box with a size of $L = 100\sigma$. The polymer system was then compressed to a small spherical confinement radius of $R_c = 20\sigma$ (or $\phi = 0.6$) using indented walls and Lennard-Jones (LJ) interactions. The force exerted by a spherical indenter on each bead is represented by the equation $F(r) = -K_{\text{indent}}(r - R_{\text{indent}})^2$, where $K_{\text{indent}}$ is the specified force constant, $r$ is the distance from

the bead to the center of the indenter, and $R_{\mathrm{indent}}$ is the radius of the indenter [83]. The intra-chain interactions were attractive, with a strength of $\epsilon_{\mathrm{intra}} = 1k_BT$ and a cutoff distance of $r_c = 2.5\sigma$, while the inter-chain interactions were repulsive, with a strength of $\epsilon_{\mathrm{inter}} = 1k_BT$ and a cut-off distance of $r_c = 2^{1/6}\sigma$. A snapshot of the initial structure is shown in a confinement with a volume fraction of $\phi = 0.1$, which illustrates that the chains were initially separated and condensed. (PDF)

**S3 Fig. Confinement diameter smaller than the radius of gyration maximizes the chromosome mixing index.** (a) In the phantom chain case ($\epsilon = 0$), the ratio of the diameter of confinement ($2R_c$) to the radius of gyration of a random-walk chain ($R_g$) is plotted as a function of the chain volume fraction in the nucleus. S5c Fig shows that the radius of gyration of a random-walk chain of $M = 8810$ beads is $R_g = 90\sigma$ when the persistence length is 5 beads. The inset of the figure shows the time-average of the chromosome mixing index ($\bar{\alpha}$) as a function of the chain volume fraction ($\phi$) in the nucleus. From both figures, it is clear that the chromosome mixing index reaches its maximum value when $2R_c/R_g < 1$. Thus when $R_g$ is large relative to $2R_c$ the confinement is effective; in the opposite case, the confinement is so large that the chains hardly mix with their diffusion in the large confinement volume greatly impeding mixing, even for phantom chains. (b) In the case of excluded volume chains (where $\epsilon \neq 0$), the confinement has an even larger effect. The radius of gyration for 4 chains of M beads each (4M beads in total, equivalent to the *Drosophila* genome) is taken to be $R_g = 180\sigma$. It is noteworthy that for $\phi = 0.01$, in the phantom chain case (a) $2Rc/Rg > 1$, but in the non-phantom chains case (b) $2Rc/Rg < 1$. (PDF)

**S4 Fig. Contact probability of phantom chain as a function of chain volume fraction $\phi$.** The contact probability of a phantom chain, which is defined as the probability of a chain's beads coming into contact in the 3D space, is shown as a function of the volume fraction $\phi$ of the chain. The contact probability follows a power law relation with the contour distance $s$ as $P_1(s) \sim s^{-\gamma}$. The exponent $\gamma$ is plotted for three different volume fractions: $\phi = 0$ (unconfined single phantom chain), $\phi = 0.001$ (4 chain simulation in relatively large confinement volume), and $\phi = 0.1$ (4 chain simulation in relatively small confinement volume). For the unconfined polymer, $\phi = 0$, the exponent is $\gamma = 1.5$, which is same to the theoretically calculated value of $\gamma = 1.5$ [39]. In a relatively large confinement volume, the exponent is $\gamma = 1.5$, and in a relatively small confinement volume, the exponent is $\gamma \approx 1.5$ between $20 < s < 200$, and saturates to a constant for $s > 200$ beads, as expected for an equilibrium polymer [87]. Note that these results are based on the contact probability $P_1(s)$ calculated for the first chain from a simulation of four phantom chains. (PDF)

**S5 Fig.** (a) Late time snapshots ($t = 10^6\tau$ time steps) of simulated, unconfined single chains for persistence lengths of $l_p = 1$ and $l_p = 5$ beads as a function of the LJ interaction strength, $\epsilon$. The chain is colored from blue to red along its length to exhibit the mixing of different segments of the chain. The radius of gyration as a function of LJ attraction strength for the persistence length (b) $l_p = 1$ beads and (c) $l_p = 5$ beads. Simulated results are compared with the scaling law of the radius of gyration ($R_g \sim M^\nu$). (d) Contact probability scaling exponent as a function of $\epsilon$ for $l_p = 1$ bead (gray color) and $l_p = 5$ bead (black color). Mean-square displacement (MSD) of a bead (averaged over all the beads of the chain) is calculated for the persistence length (e) $l_p = 1$ bead and (f) $l_p = 5$ beads. In both figures (e) and (f), the MSD fit to $\tau^{1/2}$ (black dotted line) for $\epsilon = 0$ and 0.25 and fit to $\tau^{1/4}$ (red dotted line) for $\epsilon = 0.5, 0.75, 1$. (PDF)

**S6 Fig. Second virial coefficient (SVC) of the Lennard-Jones potential.** The LJ potential describes both the attraction and repulsion between non-bonded beads. SVC is used to determine the value of $\epsilon$ for which repulsive and attractive bead-bead interactions are equal (theta solvent condition). The integral for $B(T)$ cannot be calculated analytically, but an accurate approximation can be derived from the expansion $B(T) =$

$-(2\pi/3)N_A\sigma^3 \lim_{k\to\infty}\left[\sum_{j=0}^{k}\frac{2^{(j+1/2)}}{4\,j!}\Gamma\left(\frac{2j-1}{4}\right)\left(\frac{\epsilon}{k_B T}\right)^{(2j+1)/4}\right]$ [88]. In the figure, the function $B(T)$ is plotted as a function of $\epsilon/k_B T$ for $k = 1$ to 10. For larger values of k, the graph saturates, and at $B = 0$, we get $\epsilon_\theta = 0.2925\,k_B T$.
(PDF)

**S7 Fig. Mean-square displacement (MSD) of a bead (averaged over all the beads of the chain) confined chromosome chains.** In (a),(b), and (c), MSD are calculated from simulations for phantom chains ($\epsilon = 0$), repulsive chains ($\epsilon = 0.25$), and attractive chains ($\epsilon = 0.5$) for a confinement volume equivalent to a bead volume fraction of $\phi = 0.4$. Within this confinement, the MSD of the bead increases with time, and when its value reaches the square of the confinement radius, the MSD saturates at a constant value. (a) When beads are disconnected, for $\epsilon = 0$, the MSD increases linearly with time (MSD $\sim \tau$) and for $\epsilon = 0.25$ and 0.5, MSD $\sim \tau^{0.8}$. These results demonstrate that for $\epsilon = 0$, beads behave as independent diffusive particles, and sub-diffusion occurs when we introduce interactions (repulsion or attraction) between them. MSD of chain with persistence lengths of (b) 1 bead and (c) 5 beads, for different interaction strengths, $\epsilon$, are shown. For $\epsilon = 0$, MSD $\sim \tau^{0.5}$ is the result expected from Rouse chains [15]. For repulsive ($\epsilon = 0.25$) and attractive ($\epsilon = 0.5$) interactions, MSD $\sim \tau^{0.4}$ [16]. We note that repulsion from the spherical wall speeds up the collapse of the chain compared to the unconfined case in S5f Fig. (d) MSD is calculated from our block copolymer model in the presence of lamina, for a confinement volume equivalent to a bead volume fraction of $\phi = 0.3$. This result reveals very slow mixing dynamics, as evidenced by the MSD $\sim \tau^{0.26}$ scaling law. Remarkably, the exponent of time in the MSD matches that predicted by reptation dynamics [12].
(PDF)

**S8 Fig. Mixing of four chains in a large confinement volume (corresponding to a small volume fraction of beads, $\phi = 0.001$) while varying bead-bead interactions, $\epsilon$.** (a) Simulation snapshots show the mixing of four chains mixing as a function of interaction strength, $\epsilon$, for persistence lengths $l_p = 1$ bead and $l_p = 5$ beads. Note that some snapshots were zoomed out because they were too large and would take up too much space if they were shown at their actual size. (b) The radius of gyration as a function of $\epsilon$ for $l_p = 1$ bead (gray color) and $l_p = 5$ beads (black color). Note that the attraction strength at which collapse occurs for the one-bead persistence length is $\epsilon_c = 0.3$, whereas for the 5-beads persistence length, $\epsilon_c = 0.4$.
(PDF)

**S9 Fig. Chromosome mixing index of disconnected beads as a function of time.** Figure shows the relationship between the chromosome mixing index and time in simulation runs of different durations. (a) When the simulation was run for a shorter period of time ($t = 10^3\tau$), the chromosome mixing index follows a power law relationship with time. (b) When the simulation was run for a longer period of time ($t = 10^4\tau$), the chromosome mixing index reaches a constant, maximal value ($\alpha = 3$). (c) Snapshots and calculated chromosome mixing index values from the simulation of mixing disconnected beads at various times.
(PDF)

**S10 Fig. Mixing of disconnected chromosome beads varying the bead-bead interactions, $\epsilon$.** Simulation results are shown for volume fraction $\phi = 0.4$. (a) Mixing index $\alpha$ are shown for different $\epsilon$. For $\epsilon = 0.25$, disconnected beads mix quickly and reach the maximum value of the mixing index ($\alpha = 3$). (b) Late time snapshot ($t = 10^6 \tau$ time steps) of simulated disconnected chromosome beads for attraction strength $\epsilon = 0.25$.
(PDF)

**S11 Fig. Mixing of four chains in a small confinement volume (for a relatively large volume fraction of chains, $\phi = 0.1$) varying bead-bead interactions, $\epsilon$.** (a) Simulation snapshots of the mixing of four chains as a function of interaction strength, $\epsilon$, for persistence lengths $l_p = 1$ bead and $l_p = 5$ beads. (b) The scaling exponent $\beta$ of the time dependence of the chromosome mixing index. (c) The scaling exponent $\gamma$ of the contact probability as a function of $\epsilon$ for $l_p = 1$ (gray color) and $l_p = 5$ (black color).
(PDF)

**S12 Fig. Mixing of four chains with weak attraction strength $\epsilon = 0.25$ while varying the confinement volume (equivalent to fixing the volume fraction of chains $\phi$).** Simulation snapshots of the mixing of four chains as a function of volume fraction, $\phi$, for persistence lengths $l_p = 1$ bead and $l_p = 5$ beads. Note that some snapshots were zoomed out because they were too large and would take up too much space if they were shown at their actual size.
(PDF)

**S13 Fig. Matrix of mixing exponent for different pairs of volume fractions (confinement) and self-attraction ($\phi, \epsilon$).** In the matrix, the value of the mixing exponent ($\beta$) is shown for each pair of volume fractions (confinement) and self-attraction ($\phi, \epsilon$). The matrix colors correspond to the values of $\beta$: green for $\beta > 0.12$ and red for $\beta < 0.12$. The green regions in the matrix correspond to the phase diagram in Fig 4b, which indicates that for $\beta > 0.12$, chromosomes mix within the reptation time.
(PDF)

**S14 Fig. Comparing the chromosome mixing index between different cycles during *Drosophila* embryogenesis.** The Hi-C data from the *Drosophila* genome [101] was used to compare the chromosome mixing index during the developmental stages of *Drosophila*, specifically in nuclear cycle 12 (nc12), 13 (nc13), and 14 (nc14). The comparison revealed that chromosome mixing increased progressively from nc12 to nc14.
(PDF)

**S15 Fig. Simulation comparison of inter-chromosome and intra-chromosome attraction strength.** Same versus different: (a) Late time snapshot ($t = 10^6 \tau$) for the case where the inter-chain attraction is smaller than the intra-chromosomal attraction. In this case, the chromosomes remain phase-separated in territories and are not mixed. (b) Long time snapshot ($t = 10^6 \tau$) for the case where the inter-chain attraction is equal to the intra-chain attraction; this is the physically relevant situation and as time proceeds, the chains begin to mix. (c) and (d) Contact are maps calculated from the simulation for cases (a) and (b).
(PDF)

**S16 Fig. Comparing the power law of the chromosome mixing index calculated from different types of potentials.** In the harmonic + simple Lennard-Jones (LJ) model, the bonded interaction between the chain's beads is described by a harmonic potential, while the non-bonded interaction is described by a truncated LJ potential (for more information on these potentials, see the Materials and methods section). In the FENE + simple LJ model, the bonded interaction is described by a finite extensible non-linear elastic (FENE) spring, while the non-

bonded interaction is described by a truncated LJ potential (for more information on the FENE potential, see the reference [48]). In the FENE + shifted LJ model, the bonded interaction is described by a FENE spring, while the non-bonded interaction is described by a Week-Chandler-Andersen (WCA) potential for excluded volume and a truncated and shifted LJ potential for attractions (for more information on the shifted LJ potential, see the reference [48]). In the figure, we see that the scaling law for the mixing index as a power of the time (obtained from the slope of the log-log plot) is very similar for all these potentials.
(PDF)

**S1 Table. Simulation parameters.**
(PDF)

**S2 Table. Variation of scaling exponent of MSD.**
(PDF)

**S3 Table. Comparison between scaling exponent of RMSD and scaling exponent of mixing index.**
(PDF)

**S1 Video. Simulation video of mixing of four phantom chains in small confinement ($\phi =$ 0.1) over a time period of $10^6 \tau$.**
(AVI)

**S2 Video. Simulation video of mixing of four weakly attractive chains ($\epsilon = 0.25\ k_B T$) in small confinement ($\phi = 0.1$) over a time period of $10^6 \tau$.**
(AVI)

**S3 Video. Simulation video of mixing of four attractive chains ($\epsilon = 0.5\ k_B T$) in small confinement ($\phi = 0.1$) over a time period of $10^6 \tau$.**
(AVI)

**S4 Video. Simulation video of mixing of four very strongly attractive chains ($\epsilon = 1\ k_B T$) in small confinement ($\phi = 0.1$) over a time period of $10^6 \tau$.**
(AVI)

**S5 Video. Simulation video of mixing of *Drosophila* chromosomes in presence of lamina in small confinement ($\phi = 0.3$) over a time period of $10^6 \tau$. Spherical confinement cut at the equatorial plane to show chromosome beads in one hemisphere.**
(AVI)

**S6 Video. Simulation video of mixing of disconnected weakly attractive chromosome beads ($\epsilon = 0.3\ k_B T$) in small confinement ($\phi = 0.4$) over a short time period ($t = 10^3 \tau$).**
(AVI)

## Acknowledgments

We thank Prof. Zhen-Gang Wang, Prof. Philip Pincus, Prof. Talila Volk, Omar Adame-Arana, Amit Kumar, Dana Lorber, Dan Deviri, and Shensheng Chen for useful discussions.

## Author Contributions

**Conceptualization:** Gaurav Bajpai, Samuel Safran.

**Data curation:** Gaurav Bajpai.

**Formal analysis:** Gaurav Bajpai, Samuel Safran.

**Funding acquisition:** Samuel Safran.

**Investigation:** Gaurav Bajpai, Samuel Safran.

**Methodology:** Gaurav Bajpai, Samuel Safran.

**Project administration:** Samuel Safran.

**Resources:** Samuel Safran.

**Software:** Gaurav Bajpai.

**Supervision:** Samuel Safran.

**Validation:** Gaurav Bajpai, Samuel Safran.

**Visualization:** Gaurav Bajpai, Samuel Safran.

**Writing – original draft:** Gaurav Bajpai, Samuel Safran.

**Writing – review & editing:** Gaurav Bajpai, Samuel Safran.

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
