## [Decision Letter · Decision Letter 0]

1 Dec 2022

Dear Dr. Bajpai,

Thank you very much for submitting your manuscript "Mesoscale, long-time mixing of chromosomes and its connection to polymer dynamics" for consideration at PLOS Computational Biology.

As with all papers reviewed by the journal, your manuscript was reviewed by members of the editorial board and by several independent reviewers. In light of the reviews (below this email), we would like to invite the resubmission of a significantly-revised version that takes into account the reviewers' comments.

The paper was seen and evaluated by three experts in the field. While all three referees have appreciated the motivation of the study (revealing the dynamics of inter-chromosomal mixing), they also have raised significant concerns about the presentation as well as the technical details of the manuscript. Based on their comments, it appears that substantial revisions are inevitable before any further decision is made.

We cannot make any decision about publication until we have seen the revised manuscript and your response to the reviewers' comments. Your revised manuscript is also likely to be sent to reviewers for further evaluation.

Sincerely,

Changbong Hyeon

Academic Editor

PLOS Computational Biology

Lucy Houghton

Staff

PLOS Computational Biology

The paper was seen and evaluated by three experts in the field. While all three referees have appreciated the motivation of the study (revealing the dynamics of inter-chromosomal mixing), they also have raised significant concerns about the presentation as well as the technical details of the manuscript. Based on their comments, it appears that substantial revisions are inevitable before any further decision is made.

Reviewer's Responses to Questions

**Comments to the Authors:**

Reviewer #1: The review report is uploaded as an attachment.

Reviewer #2: In this manuscript, Bajpai & Safran propose to systematically study chromosome intermingling using polymer simulations. In particular, they focus on the role of confinement and self-interaction on the dynamics of chromosome territories mixing.

Recently, in the biological community, the interest for functional inter-chromosomal contacts increased (see recent Peter Fraser’s work for example). Therefore, a generic and systematic analysis of chromosome intermingling might be of interest. In particular, since the number of modeling works addressing that question are quite rare.

However, the manuscript contains several major issues, both on substance and form, that have to be addressed or discussed.

Major comments:

- On the form:

o The manuscript is very descriptive and contains lots of unclear/unjustified statements (see other remarks below).

o It contains also many typos (a careful proofreading is required) and lots of repetitions that can be removed (eg, on the definition of P_c or beta in lots of results subsection).

o The introduction is strangely written: first there is a brief standard-like introduction reviewing the literature (but containing again many repetitions : the abstract is there (top of page 3) or L87-99 Page 3 versus L50-61 Page 2), followed by a kind of extended summary of the results that would be more appropriate in the conclusion/discussion part.

- On the methods:

o Authors may want to say that homologous chromosomes in Drosophila are paired to motivate that they only simulate 4 chromosomes (instead of 4 pairs).

o The authors use a very rigid polymer model (Kuhn length of 300 nm, L_k/sigma=10) which is certainly non-biologically relevant in vivo. More recent estimations of chromatin fiber rigidity lead to L_k=100 nm (eg, https://doi.org/10.1093/nar/gkz374). May be nice to discuss more the differences/similarities between the simulations done of l_p=5 and 1.

o Usually, FENE potentials are more adapted to simulate topologically constraint polymers. Why choosing a simple harmonic potential?

o Authors use the same Lennard-Jones potential to describe steric interactions and attractions. Again, this is non-standard and can actually be confusing. Indeed, as epsilon is changed, it modifies both the excluded volume and attraction contributions, in particular as the LJ potential is not shifted (but truncated). It might be better to use a standard Week-Chandler-Andersen potential for excluded volume and a truncated and shifted LJ potential for attractions (see eg, https://doi.org/10.1016/j.celrep.2019.08.045).

o The details about the block copolymer simulations (and LAD) are not described.

o For a more realistic time mapping between simulation and real time units, it is better to map the simulated MSD with experimental measurements (see eg https://doi.org/10.1016/j.molcel.2018.09.016) (with Fig.S1 for example).

o To calculate the predicted Hi-C data, authors use a threshold distance of 1.5 sigma = 45 nm, while Hi-C data have been associated to threshold distance of the order of 100-200 nm (eg, doi:10.1101/gr.275827.121). Does it affect the main results ?

o It is unclear how the initial configurations are constructed, in particular to force that there is no knot.

o The predicted HiC shown in the figures are very noisy, meaning that authors should compute them with more statistics (ie independent simulated trajectories)

- Authors observe a non monotonic behaviors for the exponent beta as a function of eps or phi, but give little (and argued) physical explanations for this non monotonicity.

- Extrapolation subsection: authors estimate the mixing times by imposing alpha(t)=alpha_max. But this alpha_max value has a meaning only in a highly confined system where the equilibrium state is a fully mixed state. For other situations (less confinement for example), the mixing time would be better defined as the typical time to reach the equilibrium alpha value (which is likely to be different and lower than alpha_max). Also, in the case where the mixing time is minimal (phi=0.4, eps=0.3), it may be good to launch long simulations to check if the extrapolation is correct.

- Fig.5:“we observe that heterochromatic regions do not mix..”: well experimentally pericentromeric regions do mix (look at the contacts between them in a Hi-C map, eg, https://doi.org/10.1371/journal.pgen.1008673).

- Fig.5c,f+Fig6c-d: visually it is clear that the predicted Hi-C maps poorly described the experimental one (Fig6e) even if the alpha values are similar. Any comment ? (note that to be fair the pericentromeric regions in the experimental Hi-C map should be represented [even if no data can be assigned to them]). It seems also that the predictions for gamma are not really quantitative compared to the Hi-C data in fly.

- There are now available data at different time point along the cell cycle or embryogenesis for example (Dekker lab, Blobel lab, Vaquerizas lab). Would be interesting to see the experimental dynamics of alpha extracted from these data.

- In the abstract, in the results and in the conclusion, authors state that the beta exponent is related to the MSD diffusion exponent by just observing some correspondences. While it is indeed likely that both are connected, it is unclear if it is just a coincidence or if there is more physical explanations (that are not demonstrated in the present paper) behind this connections.

Minor comments

- Some unclear statements that need to be rephrased or more detailed:

o Page 2 L27-32

o P7 L251-252: really, no knots in phantom chains? where is the evidence ?

o P7 L268 “ideal limit”: unclear.

o P7 L283-286

o P9 L320-322: and so what ? what is the connection with the rest of the paragraph? What do we learn ?

o P9 L341-345: and so what ? what is the connection with the rest of the paragraph? What do we learn ?

o P11 L378-381: this statement is true for every eps value.

o P12 L402-410

o P12 L420-421: I do not think there is evidence that a nematic ordering is observed in mitotic chromosome.

o P15 L461: VIBGYOR?

o P15 L474-483: biological relevance of such long time-scales?

- In Drosophila, an 8 min cell cycle is only observed during early embryogenesis (cycle number < 13), then cell cycles become longer (hours) and potentially cells stop dividing.

- P4 L140: regarding gel-like analysis, authors may consider https://doi.org/10.1038/s41467-019-10628-9 or doi:10.1101/gr.275827.121

- P 5, L166-168, authors may cite previous work integrating LAD at a genome-wide scale (eg, https://doi.org/10.1016/j.bpj.2021.11.936)

- Fig1b right: bar(alpha) is the time-average of alpha, but once the system reach equilibrium or since the beginning of the simulations?

- Fig.1b-1c: please use the same color code

- Fig.1c: I found very suspicious that for such a low density (0.001), the authors do not recover exactly the expected behavior for a phantom chain (gamma=1.5).

- Fig.1f-g: please add vertical and horizontal lines to help visualizing the separation between the different chromosomes.

- Fig.2b: for eps=1kT, is alpha really a power law ?

- Fig.2b & Fig.3b: the evolution of beta (as a function of eps or phi) seems quite noisy (lots of fluctuations), is it due to a lack of statistics ?

- Fig.3b, 3d: seems that there are a discontinuity between the first two black dots of the beta and gamma curves, why ?

- Does the authors account for the Rabl-like organization of chromosomes in Drosophila?

- In Drosophila, knock-down of condensin II during interphase leads to more chromosome mixing (https://doi.org/10.1371/journal.pgen.1007393), suggesting that SMC might have a impact on territories formation. May be interesting to discussed that.

Reviewer #3: The paper "Mesoscale, long-time mixing of chromosomes and its connection to polymer dynamics" by Bajpai & Safran presents results of a computational polymer model for chromosome organization.

In particular, the authors are interested in the phenomenology of chromosome mixing during interphase and, to this purpose, they study different scenarios: ideal chains (i.e., no interactions between polymers/chromosomes), interacting chains with short-range monomer-monomer attractions, polymer-lamina interactions.

In order to distinguish how polymer chains mix in these different contexts, the authors introduce a simple classifier: the mixing index \\alpha, corresponding to the ratio between the inter-chromosomal interactions and the intra-chromosomal interactions. In the intentions of the authors, the time behavior of this exponent monitors how different chromosomes mix over time.

The main topic of the paper (chromosome organization) is timely and using generic polymer models to understand it is certainly interesting. Said that, I see no element of novelty in this paper which may justify its publication in Plos Computational Biology, and I try to motivate my decision:

1) I have found the paper confusing, in particular I do not understand why the authors employ different models: ideal polymers (i.e., no excluded-volume), polymers with excluded volume in bulk, confined polymers. If the purpose of the authors is to test different models, how does all this connect to chromosomes?

2) The authors claim that the mixing index \\alpha may be used to quantify chromosome mixing within the cells: I find this argument particularly weak, because - with respect to the simulations presented by the authors - \\alpha depends 'a lot' from parameters like polymer density, polymer overlap in solution, etc... Given the great variability across different cells and different organisms, it is hard to believe that \\alpha gives any new insight on chromosomes.

3) I have found the presentation of the work poorly organized. There are frequent repetitions in the text (for instance, in the Introduction, the sentence starting at line 57 and the sentence starting at line 93 are basically the same), not to mention that there are too many frequent typos as "the the" in the caption of Fig. 3, line 8 which indicate that the text has not been proofread duly.

4) Last but not least, the authors dedicate a whole paragraph to 'Physical units of chromosome mixing models' , in particular they say that their time units corresponds to 60us in 'real time'. This looks ok, but why then - throughout the text - do they seem to forget about it and continue referring to MD time units \\tau? They should have used 'real time' units everywhere.

In conclusion, it is my opinion that this work does not contain material suitable for publication in Plos CB. In addition, the level of presentation is not adequate for publication and, should the authors decide to resubmit elsewhere, careful proofreading of the text is urged.

**Have the authors made all data and (if applicable) computational code underlying the findings in their manuscript fully available?**

Reviewer #1: Yes

Reviewer #2: **No: **simulation code (LAMMPS input files) are not available

Reviewer #3: Yes

PLOS authors have the option to publish the peer review history of their article (what does this mean?). If published, this will include your full peer review and any attached files.

Reviewer #1: No

Reviewer #2: No

Reviewer #3: No
---

## [Decision Letter · Decision Letter 1]

19 Feb 2023

Dear Dr. Bajpai,

Thank you very much for submitting your manuscript "Mesoscale, long-time mixing of chromosomes and its connection to polymer dynamics" for consideration at PLOS Computational Biology. As with all papers reviewed by the journal, your manuscript was reviewed by members of the editorial board and by several independent reviewers. The reviewers appreciated the attention to an important topic. Based on the reviews, we are likely to accept this manuscript for publication, providing that you modify the manuscript according to the review recommendations.

While the reviewers #1 and #3 concluded the suitability of the revised manuscript for publication, the reviewer #2 is still raising substantive concerns.

Overall, the content of the paper appears interesting, and I believe that the paper is worth publishing in PLoS Comp. Biol.; however, the authors should address the referee's comments more thoroughly prior to it.

Sincerely,

Changbong Hyeon

Academic Editor

PLOS Computational Biology

Lucy Houghton

Staff

PLOS Computational Biology

While the reviewers #1 and #3 concluded the suitability of the revised manuscript for publication, the reviewer #2 is still raising substantive concerns.

Overall, the content of the paper appears interesting, and I believe that the paper is worth publishing in PLoS Comp. Biol.; however, the authors should address the referee's comments more thoroughly prior to it.

Reviewer's Responses to Questions

**Comments to the Authors:**

Reviewer #1: All my questions have been well answered and the manuscript has been improved a lot. I am satisfied with the current version.

Reviewer #2: Authors have addressed some of my previous concerns but eluded many and, while the subject of study is potentially interesting, the revised manuscript still does not meet the standards of a comprehensive, complete paper that can be published in PLoS CB.

Regarding author answers to my previous comments:

- The authors still have not done a proper time mapping. With their estimation, they find \\tau=60 us. Looking at FigS11, a MSD of 100 (bead unit)^2 (that would correspond to about 0.1 um^2) is achieved after about 10^4 \\tau ~ 0.6 sec from their estimation. While in yeast but also in other species a MSD of 0.1 um^2 is obtained at time of the order of 100 sec !

- On the fact that Hi-C map are very noisy: I do not see why the use of a minimal model may be the cause of this “noise”. The four polymers are strictly identical so their average intra-chromosome contact maps should be similar and the inter-chromosome patterns also. In all the shown Hi-C maps, intra and inter- patterns are very different. It is not acceptable to make solid conclusions with such poor statistics. I do understand that it may take time but this is required (and I do not see why “scaling laws are often robust and not sensitive to the many microscopic details” may change that). Or maybe one possibility is to correctly compute/estimate error bars that accounts for the lack of statistics. For example, in Fig.2 c, visually it seems that the intra-chromosomal contact are stronger for eps=0.5 than for eps=1, while on Fig.2d P(s) for eps =1 is higher than for eps=0.4. How is it possible ? Idem for Fig.3c,d

- On the fact that in the phantom chains, \\gamma=1.25 for phi=0.001. According to their computation for phi=0.001, the radius of gyration of the chain is lower than the sphere diameter. Thus, the effect of confinement should be limited and thus one would expect an exponent of -1.5 almost over the whole range of genomic distance. Even in the globular case (strong confinement), the first part of the P(s) curve (short genomic distance) towards the constant exponent (FigS5) should have an exponent of -1.5 (see Mirny, Chromosome research, 2011).

- On the mixing of pericentromeric domains: experimentally, PCH domains do mix while in authors’ model they don’t. This should be discussed.

- On the comparison between predicted and experimental Hi-C maps: claiming that both are comparable is not justified (look at Fig 5 & 6), and this should be discussed in the text more deeply. Claiming that in Hi-C experiment, the radius of the nucleus is unknown is wrong. There certainly exist estimation of the size of the nucleus in late embryos and so authors may estimate a Phi value.

- On the changes of alpha during embryogenesis: I was suggesting some recent datasets: in Drosophila the one from Vaquerizas is of excellent quality (Cavalli lab also produced some). And it’s clear from these data that chromosomes become more and more mixed as embryogenesis progresses (and not the reverse).

New issues regarding the added text:

- On the form:

o The paper is still sometimes hard to grasp (and still contains a non-negligible number of typos) (even for a specialist). Non-expert researchers interested in chromosome organization may still have hard times to read it.

o The introduction is still contained an extended summary of the results that would be more appropriate in the conclusion/discussion part (from Page 3 Line73 to Page 4 L157).

o The revised manuscript still contains many unclear, unjustified or even wrong statement.

As I suggested in my first review, authors added some references to effect of condensin II or for homologous pairing. But did not discuss them (because they highlight some limitations of the study) or introduce them wrongly.

• For example, they cite the condensin II observation in the abstract and also elsewhere in the text, while indeed it may suggest that mixing is under control, it is via a mechanism (loop extrusion) that is not integrated in the present paper. By the way, the sentence “we do not include the short-time ….noise” (L45 Page 2) that is supposed to justify to not integrate loop extrusion is incorrect. Short-time dynamics mechanism (like loop extrusion) may have deep, large-scale effect on genome folding and may not be neglected ! This should be clearly (and fairly) discussed in the Discussion.

• Regarding homologous pairing, this is not because “homologous chromosomes share significant sequence similarity that they generally behave similarly” (Page 5 L 159) but in drosophila homologous chromosomes are paired at the molecular scale (btw this may also potentially affect the mixing).

Page 2 L20-25: unclear sentence

Page 3 L 88: “we note that all these are structural and not dynamical characterization”: actually there exist many (theoretical) studies of chromatin dynamics (eg Ref 36, Ref 41, Ref2, Ref 58).

Page 20 L627-637: unclear

Page 16 L510-516: unclear.

- I appreciate the new simulations to show that using FENE or truncated & shifted potentials results may be equivalent. But I cannot see reference into the main text to the corresponding Sup Fig.

- In all figures showing simulation snapshots, showing the spherical confinement would allow to visualize better the organization within the sphere (+ adding a ruler to give information on distances)

- Page 20 L 614-616: “previous studies that simulated…used different interactions for intra-….”: well most hypothesis driven models do not assume different interactions but same energy strengths (including those in Ref 19 &27).

- Page 14 L476-479: “greater than the typical cell cycle…” : indeed but lower than the typical cell cycle in late embryos…

Reviewer #3: With the present version of the paper, the authors provide a significantly reviewed version of their work.

Upon the first submission, due to a non-adequate presentation of both methods and results, I had the impression that the paper was not suitable for publication in Plos CB. The present version is significantly different from the first, and much more convincing. I am particularly pleased to notice that the authors made a serious effort to address both my reservations and those by the other reviewers.

For these reasons, I believe that the paper has now the appropriate form for publication in Plos CB.

**Have the authors made all data and (if applicable) computational code underlying the findings in their manuscript fully available?**

Reviewer #1: Yes

Reviewer #2: **No: **

Reviewer #3: Yes

PLOS authors have the option to publish the peer review history of their article (what does this mean?). If published, this will include your full peer review and any attached files.

Reviewer #1: No

Reviewer #2: No

Reviewer #3: No

Figure Files:

Data Requirements:

Reproducibility:

References:

---

## [Editor Report · Decision Letter 2]

1 May 2023

Dear Dr. Bajpai,

We are pleased to inform you that your manuscript 'Mesoscale, long-time mixing of chromosomes and its connection to polymer dynamics' has been provisionally accepted for publication in PLOS Computational Biology.

Best regards,

Changbong Hyeon

Academic Editor

PLOS Computational Biology

Lucy Houghton

Staff

PLOS Computational Biology

The paper has been revised constructively addressing the reviewer #2's comments. I therefore recommend its acceptance for the publication in PLOS Comp. Biol.

---

## [Editor Report · Acceptance letter]

19 May 2023

PCOMPBIOL-D-22-01474R2 

Mesoscale, long-time mixing of chromosomes and its connection to polymer dynamics

Dear Dr Bajpai,

I am pleased to inform you that your manuscript has been formally accepted for publication in PLOS Computational Biology. Your manuscript is now with our production department and you will be notified of the publication date in due course.

With kind regards,

Anita Estes
